# DNA sequence encodes the position of DNA supercoils

**Sung Hyun Kim[†‡], Mahipal Ganji[†§#], Eugene Kim, Jaco van der Torre, Elio Abbondanzieri[¶|]\*, Cees Dekker\***

Department of Bionanoscience, Kavli Institute of Nanoscience, Delft University of Technology, Delft, The Netherlands

**\*For correspondence:**
elio.abbondanzieri@rochester.edu (EA);
C.Dekker@tudelft.nl (CD)

[†]These authors contributed equally to this work

**Present address:** [‡]Institute of molecular biology and genetics, School of Biological Science, Seoul National University, Seoul, South Korea; [§]Department of Physics, Center for Nanoscience, Ludwig Maximilian University, Munich, Germany; [#]Max Planck Institute of Biochemistry, Martinsried, Germany; [¶]Department of Biology, University of Rochester, New York, United States

**Competing interests:** The authors declare that no competing interests exist.

**Abstract** The three-dimensional organization of DNA is increasingly understood to play a decisive role in vital cellular processes. Many studies focus on the role of DNA-packaging proteins, crowding, and confinement in arranging chromatin, but structural information might also be directly encoded in bare DNA itself. Here, we visualize plectonemes (extended intertwined DNA structures formed upon supercoiling) on individual DNA molecules. Remarkably, our experiments show that the DNA sequence directly encodes the structure of supercoiled DNA by pinning plectonemes at specific sequences. We develop a physical model that predicts that sequence-dependent intrinsic curvature is the key determinant of pinning strength and demonstrate this simple model provides very good agreement with the data. Analysis of several prokaryotic genomes indicates that plectonemes localize directly upstream of promoters, which we experimentally confirm for selected promotor sequences. Our findings reveal a hidden code in the genome that helps to spatially organize the chromosomal DNA.
DOI: https://doi.org/10.7554/eLife.36557.001

## Introduction

Control of DNA supercoiling is of vital importance to cells. Torsional strain imposed by DNA-processing enzymes induces supercoiling of DNA, which triggers large structural rearrangements through the formation of plectonemes (*Vinograd et al., 1965*). Recent biochemical studies suggest that supercoiling plays an important role in the regulation of gene expression in both prokaryotes (*Le et al., 2013*) and eukaryotes (*Naughton et al., 2013*; *Pasi and Lavery, 2016*). In order to tailor the degree of supercoiling around specific genes, chromatin is organized into independent topological domains with varying degrees of torsional strain (*Naughton et al., 2013*; *Sinden and Pettijohn, 1981*). Domains that contain highly transcribed genes are generally underwound whereas inactive genes are overwound (*Kouzine et al., 2013*). Furthermore, transcription of a gene transiently alters the local supercoiling (*Kouzine et al., 2013*; *Naughton et al., 2013*; *Peter et al., 2004*), while, in turn, torsional strain influences the rate of transcription (*Chong et al., 2014*; *Liu and Wang, 1987*; *Ma et al., 2013*).

For many years, the effect of DNA supercoiling on various cellular processes has mainly been understood as a torsional stress that enzymes should overcome or exploit for their function. More recently, supercoiling has been acknowledged as a key component of the spatial architecture of the genome (*de Wit and de Laat, 2012*; *Dekker et al., 2013*; *Ding et al., 2014*; *Neuman, 2010*). Here, bound proteins are typically viewed as the primary determinant of sequence-specific tertiary structures while intrinsic mechanical features of the DNA are often ignored. However, the DNA sequence influences its local mechanical properties such as bending stiffness, curvature, and duplex stability, which in turn alter the energetics of plectoneme formation at specific sequences (*Dittmore et al., 2017*; *Irobalieva et al., 2015*; *Matek et al., 2015*). Unfortunately, the relative importance of these factors that influence the precise tertiary structure of supercoiled DNA have remained unclear

(*Dekker and Heard, 2015*). Various indications that the plectonemic structure of DNA can be influenced by the sequence were obtained from biochemical and structural studies (*Kremer et al., 1993*; *Laundon and Griffith, 1988*; *Pfannschmidt and Langowski, 1998*; *Tsen and Levene, 1997*) as well as from work performed in silico (*Eslami-Mossallam et al., 2016*; *Pasi and Lavery, 2016*; *Wang et al., 2017*). These studies suggested that plectonemes may get localized to highly curved or flexible segments of DNA. However, this examined only a handful of specific sequences such as phased poly(A)-tracts and a particular high–curvature sequence rich in poly(A)-tracts, making it difficult to determine if curvature, long poly(A)-tracts, or some other DNA feature drives the sequence–structure relationship.

Here, we study how DNA sequence governs the structure of supercoiled DNA by use of a recently developed single-molecule technique termed ISD (intercalation-induced supercoiling of DNA) (*Ganji et al., 2016b*), which uses intercalating dyes to induce supercoiling as well as to observe the resultant tertiary structures in many DNA molecules in parallel. Plectonemes are directly observable as intensity maxima along the DNA, from which their position along DNA can be

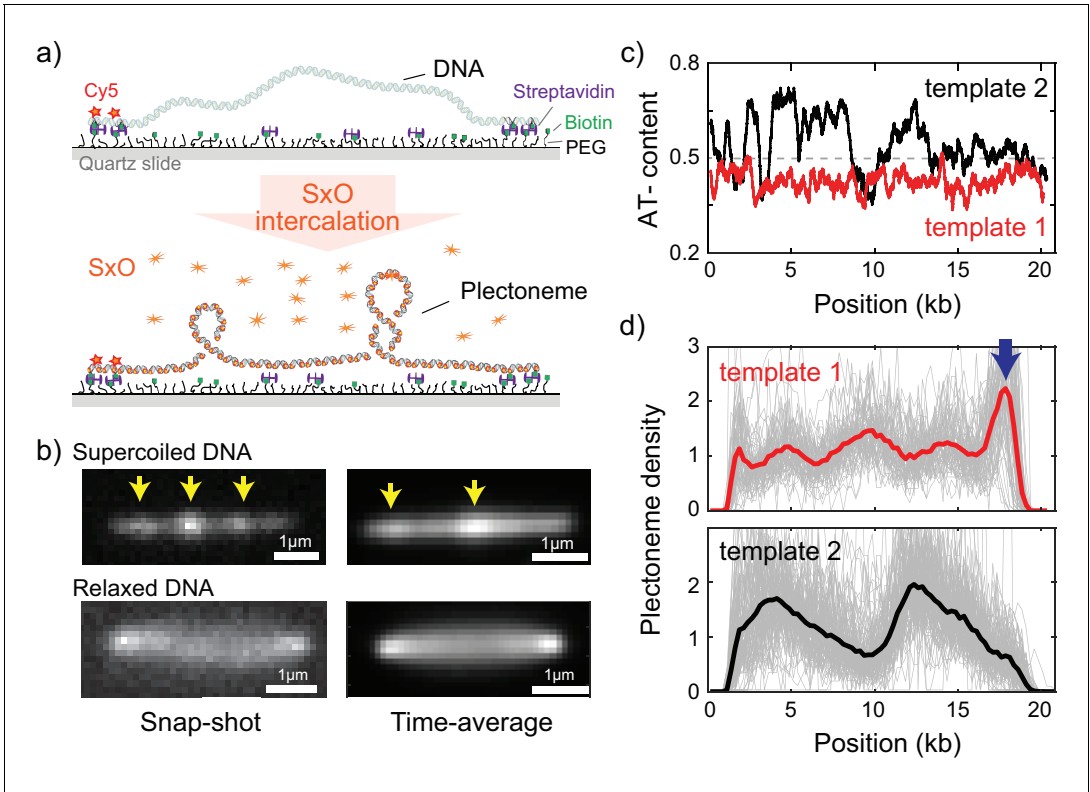

**Figure 1.** Direct visualization of individual plectonemes on supercoiled DNA. (**a**) Schematic of the ISD assay. (top) A flow-stretched DNA is doubly-tethered on a PEG-coated surface via streptavidin-biotin linkage. One-end of the DNA is labeled with Cy5-fluorophores (red stars) for identifying the direction of each DNA molecule. (bottom) Binding of SxO fluorophores induces supercoiling to the torsionally constrained DNA molecule. (**b**) Representative fluorescence images of a supercoiled DNA molecule. Left: Snap-shot image of a supercoiled DNA with 100 ms exposure. Yellow arrows highlight higher DNA density, that is individual plectonemes. Right: Time-averaged DNA image by stacking 1000 images (of 100 ms exposure each). Arrows indicate peaks in the inhomogeneous average density of plectonemes. (**c**) AT-contents of two DNA samples: template1 and template2 binned to 300 bp. (**d**) Plectoneme densities obtained from individual DNA molecules. (top) Plectoneme density on template1 (grey thin lines, n = 70) and their ensemble average (red line). Arrow indicates a strong plectoneme pinning site. (bottom) Plectoneme densities obtained from individual DNA molecules of template2 (grey thin lines, n = 120) and their ensemble average (black line).

DOI: https://doi.org/10.7554/eLife.36557.002

The following figure supplements are available for figure 1:

**Figure supplement 1.** Details of the ISD assay.

DOI: https://doi.org/10.7554/eLife.36557.003

**Figure supplement 2.** Characterization of the supercoiled and torsionally relaxed DNA.

DOI: https://doi.org/10.7554/eLife.36557.004

extracted (see *Figure 1a* and *Figure 1—figure supplement 1*). We find a strong relationship between sequence and plectoneme localization. By examining many different sequences, we systematically rule out several possible mechanisms of the observed sequence dependence. Using a model built on basic physics, we show that the local intrinsic curvature determines the relative plectoneme stability at different sequences. Application of this model to sequenced genomes reveals a clear biological relevance, as we identify a class of plectonemic hot spots that localize upstream of prokaryotic promoters. Subsequently, we confirm that these sequences pin plectonemes in our single-molecule assay, testifying to the predictive power of our model. We also discuss several eukaryotic genomes where plectonemes are localized near promoters with a spacing consistent with nucleosome positioning. Taken together, our experimental results and our physical model show a clear sequence-supercoiling relationship and indicate that genomic DNA encodes information for positioning of plectonemes, likely to regulate gene expression and contribute to the three-dimensional spatial ordering of the genome.

## Results

### Single-molecule visualization of individual plectonemes along supercoiled DNA

To study the behavior of individual plectonemes on various DNA sequences, we prepared 20 kb-long DNA molecules of which the end regions (~500 bp) were labelled with multiple biotins for surface immobilization (*Figure 1—figure supplement 1a–b*). The DNA molecule were flowed into streptavidin-coated sample chamber at a constant flow rate to obtain stretched double-tethered DNA molecules (*Figure 1a* and *Figure 1—figure supplement 1a*). We then induced supercoiling by adding an intercalating dye, Sytox Orange (SxO), into the chamber and imaged individual plectonemes formed on the supercoiled DNA molecules. Notably, SxO does not have any considerable effect on the mechanical properties of DNA under our experimental conditions (*Ganji et al., 2016b*).

Consistent with previous studies (*Ganji et al., 2016b*; *van Loenhout et al., 2012*), we observed dynamic spots along the supercoiled DNA molecule (highlighted with arrows in *Figure 1b*-top left and *Video 1*). These spots disappeared when DNA torsionally relaxed upon photo-induced nicking (*Figure 1b*-bottom left) (*Ganji et al., 2016b*), confirming that the spots were plectonemes induced by the supercoiling. Interestingly, the time-averaged fluorescence intensities of the supercoiled DNA were *not* homogeneously distributed along the molecule (*Figure 1b*-top right), establishing that plectoneme occurrence is position dependent. In contrast, torsionally relaxed (nicked) DNA displayed a featureless homogenous time-averaged fluorescence intensity (*Figure 1b*-bottom right).

### DNA sequence favors plectoneme localization at certain spots along supercoiled DNA

Upon observing the inhomogeneous fluorescence distribution along the supercoiled DNA, we sought to understand if the average plectoneme position is dependent on the underlying DNA sequence. We prepared two DNA samples; the first contained a uniform distribution of AT-bases while the second contained a strongly heterogeneous distribution of AT-bases (*Figure 1c*, template1 and template2, respectively). In order to quantitatively analyze the plectoneme distribution, we counted the average number of plectonemes over time at each position on the DNA molecules and built a position-dependent probability density function of the plectoneme occurrence (from now onwards called plectoneme density; see Materials and methods for details). The plectoneme density is normalized to its average value across the DNA such that a density value above one indicates that the region is a favorable position for plectonemes relative to other regions within the DNA molecule. For both DNA samples, we observed a strongly position-dependent plectoneme density (*Figure 1d*). Strikingly, the plectoneme densities (*Figure 1d*) were very different for the two DNA samples. This difference demonstrates that plectoneme positioning is directed by the underlying DNA sequence. Note that we did not observe any position dependence in the intensity profiles when the DNA was torsionally relaxed, indicating that the interaction of dye is not responsible for the dependence (*Figure 1—figure supplement 2a*).

The plectoneme kinetics showed a similar sequence dependence, as the number of events for nucleation and termination of plectonemes was also found to be position dependent with very

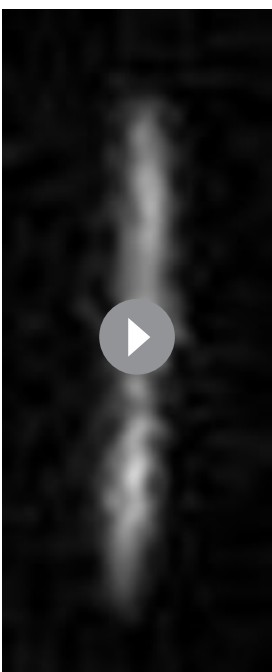

**Video 1.** A representative real-time fluorescence image of a supercoiled DNA molecule that shows dynamic bright spots upon plectoneme formation. At 20 s after acquisition, the DNA was torsionally relaxed due to photo-induced nicking.

DOI: https://doi.org/10.7554/eLife.36557.005

different profiles for each DNA samples (*Figure 1—figure supplement 2b*). Importantly, at each position of the DNA, the number of nucleation and termination events were the same, showing that the system was at equilibrium. Because the aim of our study is to examine the sequence–structure relationship in supercoiled DNA, which is an equilibrium property, we focus on analyzing the plectoneme density profiles for a variety of sequences.

## Systematic examination of plectoneme pinning at various putative DNA sequences

We first considered a number of potential links between DNA sequence and plectoneme density. Note that in particular the sharply bent apical tips of plectonemes (*Figure 1A*) create an energy barrier to plectoneme formation. This barrier could be reduced if the DNA was able to locally melt or kink, if a specific region of DNA was more flexible than others, or if the DNA sequence was intrinsically curved already before the plectoneme formed. Because all of these properties (duplex stability, flexibility, and curvature) are influenced by the AT-content, we first examined the relationship between AT-content and the measured plectoneme densities in *Figure 1c–d*. Indeed, the plectoneme density showed a weak correlation with the local AT-percentage (R = 0.33, *Figure 2—figure supplement 1a*).

In order to unambiguously link changes in plectoneme density to specific sequences of arbitrary size, we developed an assay where we inserted various short DNA segments carrying particular sequences of interest in the middle of the homogeneous template1 (*Figure 2a* and *Figure 2—figure supplement 1b*). This allowed us to easily determine the influence of the inserted sequence on plectoneme formation by measuring changes in the plectoneme density at the insert relative to the rest of the DNA strand. We examined three different AT-rich inserts: seqA, seqB, and seqC with ~60%,~65%, and ~60% AT, respectively (*Figure 2a*). Interestingly, all three samples showed a peak in the plectoneme density at the position of insertion, further supporting the idea that AT-rich sequences are preferred positions for plectonemes (*Figure 2b*). Furthermore, when we shortened or lengthened one AT-rich sequence (seqA), we found that the probability of plectoneme pinning (i.e. the area under the peak) scaled with the length of the AT-rich fragment (*Figure 2—figure supplement 1b–e*). Overall, these results suggest that plectoneme preferentially form at AT-rich regions.

However, it is clear that AT-content alone cannot be the only factor that sets the plectoneme pinning. For example, the right-end of template1 exhibits a region that pins plectonemes strongly (*Figure 1d*-top, arrow), even though this region is not particularly AT-rich (*Figure 1c*). When we inserted a 1 kb copy of this pinning region into the middle of template1 (*Figure 2c*, 'seqCopy'), we observed an additional peak in plectoneme density (*Figure 2d*, green). Given that this region had the same total AT-content as the surrounding DNA, we hypothesized that the particular distribution of A and T bases may be more important than the total AT-content alone. In particular, poly(A)-tracts influence the local mechanical properties of DNA and might be responsible for the plectoneme pinning, as suggested by early studies (*Kremer et al., 1993*; *Pfannschmidt and Langowski, 1998*; *Tsen and Levene, 1997*). To test this, we removed all poly(A) tracts of length four or higher by replacing alternative A-bases with G or C-bases in seqCopy (*Figure 2c*, 'A-G mutation'). Upon this change, the peak in the plectoneme density indeed disappeared (*Figure 2d*, blue). However, when we instead disrupted the poly(A)≥4 tracts by replacing them with alternating AT-stretches

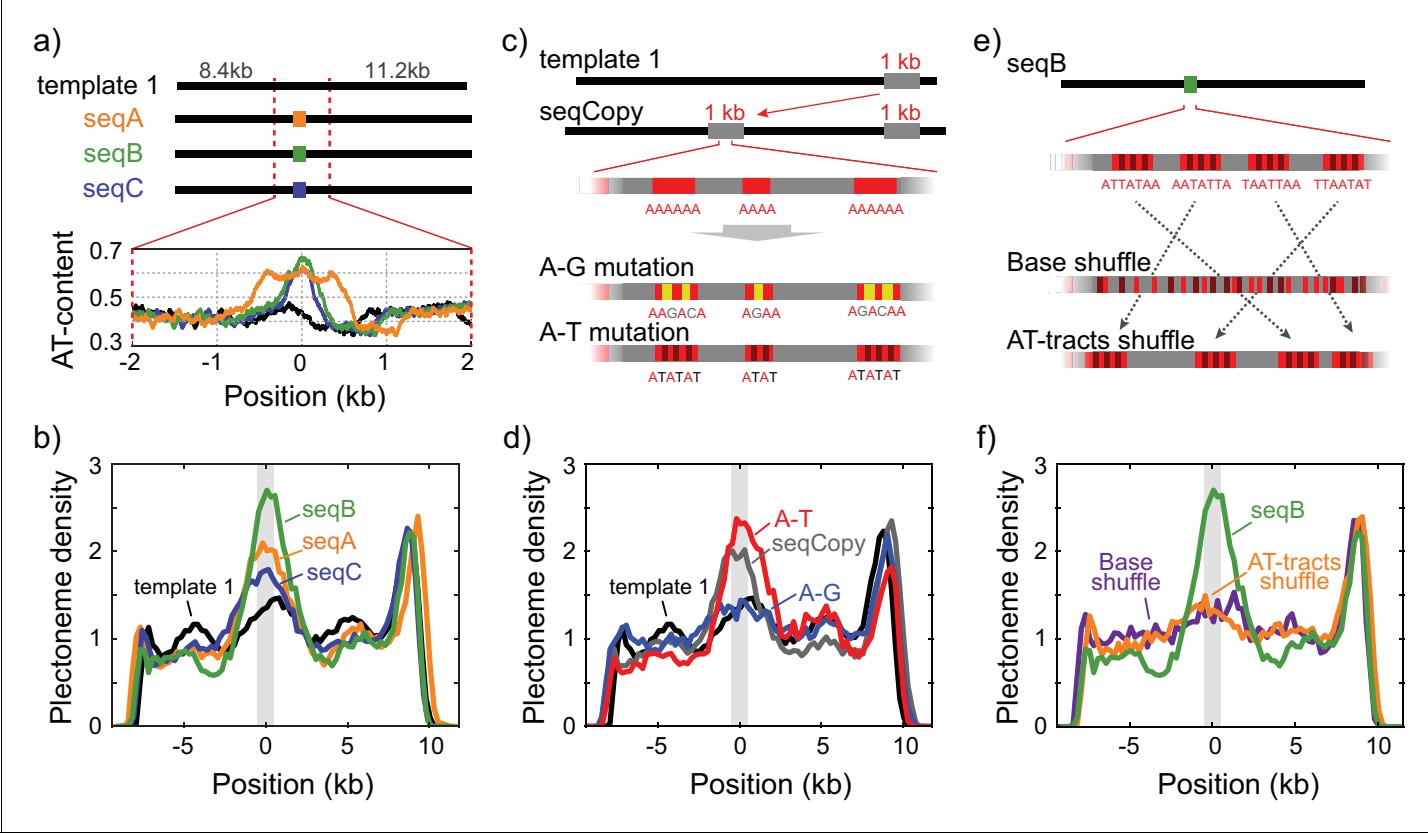

**Figure 2.** Sequence-dependent pinning of DNA plectonemes. (**a**) Top: Schematics showing DNA constructs with AT-rich fragments inserted in template1. Three different AT-rich segments, SeqA (400 bp), SeqB (500 bp), and SeqC (1 kb), are inserted at 8.8 kb from Cy5-end in template1. Bottom: AT-contents of these DNA constructs zoomed in at the position of insertion. (**b**) Averaged plectoneme densities measured for the AT-rich fragments denoted in (**A**). The insertion region is highlighted with a gray box. (n = 43, 31, and 42 for SeqA, SeqB, and SeqC, respectively) (**c**) Schematics of DNA constructs with a copy of the 1 kb region near the right end of template1 where strong plectoneme pinning is observed (seqCopy). Poly(A)-tracts within the copied region are then mutated either by replacing A bases with G or C (A-G mutation), or with T (A-T mutation). (**d**) Plectoneme densities measured for the sequences denoted in (**c**). Plectoneme density of template1 is shown in black, seqCopy in green, A-G mutation in blue, and A-T mutation in red. (n = 45, 34, and 42 for seqCopy, A-G mutation, and A-T mutation, respectively) (**e**) Schematics of DNA constructs with mixed A/T stretches modified from seqB. The insert is modified either by shuffling nucleotides within the insert to destroy all the poly(A) and poly(A/T)-tracts (Base shuffle), or by re-positioning the poly(A) or poly(A/T)-tracts (AT-tracts shuffle) – both while maintaining the exact same AT content across the insert. (**f**) Plectoneme densities measured for the sequences denoted in (**e**). seqB from panel (**b**) is plotted in green; base shuffle data are denoted in purple; AT-tracts shuffle in orange. (n = 24, and 26 for Base shuffle, and AT-tracts shuffle, respectively).

DOI: https://doi.org/10.7554/eLife.36557.006

The following figure supplement is available for figure 2:

**Figure supplement 1.** Plectoneme pinning at AT-rich inserts of various lengths.

DOI: https://doi.org/10.7554/eLife.36557.007

(*Figure 2c*, 'A-T mutation'), we, surprisingly, *did* observe strong pinning (*Figure 2d*, red), establishing that plectoneme pinning does not strictly require poly(A)-tracts either. Hence, instead of poly(A)-tracts, it could be possible that stretches consisting of either A and T ('poly(A/T)-tracts') induce the plectoneme pinning. To test this hypothesis, we re-examined the seqB construct to test if long stretches of 'weak' bases (i.e. A or T) were the source of pinning. Here, we broke up all poly(A/T)$\geq$4 tracts (i.e. all linear stretches with a random mixture of A or T bases but no G or C bases) by shuffling bases within the seqB insert while keeping the overall AT-content the same. This eliminated plectoneme-pinning, consistent with the idea that poly(A/T) tracts were the cause (*Figure 2e–f*, purple). However, if we instead kept all poly(A/T)$\geq$4 tracts intact, but merely rearranged their positions within the seqB insert (again keeping AT-content the same), this rearrangement abolished the pinning pattern (*Figure 2f*, orange), indicating that plectoneme pinning is not solely dependent on the

presence of poly(A/T) stretches, but instead is dependent on the relative positions of these stretches.

Taken together, this systematic exploration of various sequences showed that although pinning correlates with AT-content, we cannot attribute this correlation to AT-content alone, to poly(A)-tracts, or to poly(A/T)-tracts. Our data instead suggest that plectoneme pinning depends on a local mechanical property arising from the combined effect of the entire base sequences in a local region, and our shuffled poly(A/T) constructs suggest this property must be measured over distances greater than tens of nucleotides. Among the three mechanical properties we first considered, duplex stability, flexibility, and curvature, the duplex stability is unlikely to be a determinant factor for the plectoneme pinning because duplex stability is mostly determined by the overall AT/GC percentage rather than the specific distribution of bases in the local region.

## Intrinsic local DNA curvature determines the pinning of supercoiled plectonemes

To obtain a more fundamental understanding of the sequence specificity underlying the plectoneme pinning, we developed a novel physical model based on intrinsic curvature and flexibility for estimating the plectoneme energetics (see Materials and methods for details). Notably, the major energy cost for making a plectoneme is spent in inducing a strong bend within the DNA in the plectoneme tip region. Our model estimates the energy cost associated with bending the DNA into the highly curved (~240° arc) plectoneme tip (*Marko and Neukirch, 2012*). For example, at 3pN of tension (characteristic for our stretched DNA molecules), the estimated size of the bent tip is 73 bp, and the energy required to bend it by 240° is very sizeable,~18 $_k$BT (*Figure 3a–b*). However, if a sequence has a high local intrinsic curvature or flexibility, this energy cost decreases significantly. For example, an intrinsic curvature of 60° between the two ends of a 73 bp segment would lower the bending energy by a sizable amount,~8 $k_B$T. Hence, we expect that this energy difference drives plectoneme tips to pin at specific sequences. We calculated local intrinsic curvatures at each segment along a relaxed DNA molecule using published dinucleotide parameters for tilt/roll/twist (*Figure 3a* and *supplementary file 1*) (*Balasubramanian et al., 2009*). The local flexibility of the DNA was estimated by summing the dinucleotide covariance matrices for tilt and roll (*Lankas et al., 2003*) over the length of the loop. Using this approach, we estimate the bending energy of a plectoneme tip centered at each nucleotide along a given sequence (*Figure 3b*). The predicted energy landscape is found to be rough with a standard deviation of about ~1 $k_B$T, in agreement with a previous experimental estimate based on plectoneme diffusion rates (*van Loenhout et al., 2012*). We then used these bending energies to assign Boltzmann-weighted probabilities, $P_B = exp\left(-\frac{E_{loop}}{k_B T}\right)$, for plectoneme tips centered at each base on a DNA sequence. This provided theoretically estimated plectoneme densities as a function of DNA sequence. Note that we obtained these profiles without any adjustable fitting parameters as the tilt/roll/twist and flexibility values were determined by dinucleotide parameters adopted from published literature. Although both intrinsic curvature and flexibility were included, the model predicts that the flexibility is unimportant and that intrinsic curvature clearly is the dominant factor in positioning plectonemes (*Figure 3c*).

The predicted plectoneme densities (*Figure 3d* and *Figure 3—figure supplement 1*) are generally found to be in very good agreement with the measured plectoneme densities. For example, the non-intuitive mutant sequences tested above (A-G and A-T mutations) are faithfully predicted by the model (*Figure 2d* and *Figure 3d*). More generally, we find that the model qualitatively represented the experimental data for the large majority of the sequences that were tested (*Figure 3—figure supplement 1*). The simplicity of the model and the lack of fitting parameters make this agreement all the more striking. Only occasionally, we find that the model is too conservative, that is while it performs well in avoiding false positives, it suffers from some false negatives (*Figure 3—figure supplement 1*, SeqA, SeqB, and SeqC), possibly because of an insufficient accuracy in the dinucleotide parameters that we adopted from the literature. For example, different dinucleotide parameter sets from the currently available literature produce variations in the model predictions (*Figure 3—figure supplement 2*). Alternative explanations for the false negatives are also possible, for example that the local curvature is influenced by interactions spanning beyond nearest-neighbor nucleotides, or some unknown DNA sequences that stabilize twist rather than strand writhing or that are prone to base-flipping even in the positive supercoiling regime.

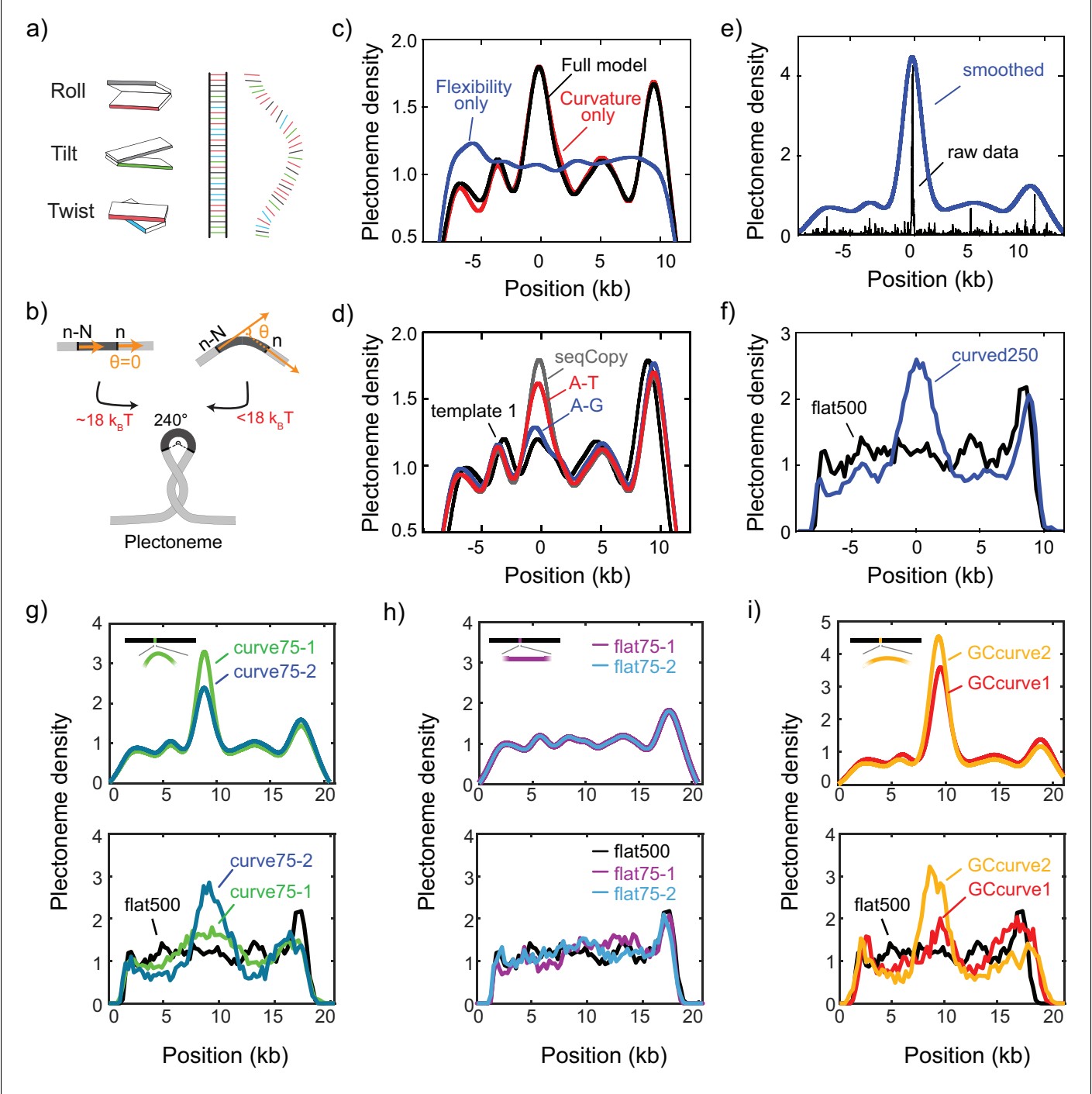

**Figure 3.** DNA plectonemes pin to sequences that exhibit local curvature. (a) Ingredients for an intrinsic-curvature model that is strictly based on dinucleotide stacking. (Left) Cartoons showing the relative alignment between the stacked bases which are characterized by three modes: roll, tilt, and twist. (Middle) In the absence of variations in the roll, tilt, and twist, a DNA molecule adopts a strictly linear conformation in 3D space. (Right) Example of a curved free path of DNA that is determined by the slightly different values for intrinsic roll, tilt, and twist angles for every dinucleotide. (b) Schematics showing the energy required to bend a rigid elastic rod as a simple model for the tip of a DNA plectoneme. (c) Plectoneme density prediction based on intrinsic curvature and/or flexibility for seqCopy. Predicted plectoneme densities calculated based on either DNA flexibility (blue), only curvature (red), or both (black). Combining flexibility and curvature did not significantly improve the prediction comparing to that solely based on DNA curvature. (d) Predicted plectoneme densities for the DNA constructs carrying a copy of the end peak and its mutations, as in *Figure 2b*. Note the excellent correspondence to the experimental data in *Figure 2b*. (e–f) Predicted (e) and measured (f) plectoneme density of a synthetic sequence (250 bp) that is designed to strongly pin a plectoneme. Raw data from the model are shown in black and its Gaussian-smoothed (FWHM = 1600 bp) is

*Figure 3 continued on next page*

*Figure 3 continued*

shown in blue in the left panel. Plectoneme densities measured from individual DNA molecules carrying the synthetic sequence (thin grey lines) and their averages (thick blue line) are shown in the right panel. (n = 37, and 21 for curved250, and flat500, respectively) (**g–h**) Model-predicted (upper panels) and experimentally measured (bottom panels) plectoneme densities of 75 bp-long highly curved (**g**) and flat (**h**) inserts. (**i**) Model-predicted (upper panels) and experimentally measured (bottom panels) plectoneme densities of curved GC-rich sequences. (n = 36, 26, 21, 20, 52, and 29 for curve75-1, curve75-2, flat75-1, flat75-2, GCcurve1, and GCcurve2, respectively).
DOI: https://doi.org/10.7554/eLife.36557.008

The following figure supplements are available for figure 3:

**Figure supplement 1.** Model-predicted plectoneme density of various sequences.
DOI: https://doi.org/10.7554/eLife.36557.009

**Figure supplement 2.** Comparison of the model predictions on seqCopy for various sets of model parameters.
DOI: https://doi.org/10.7554/eLife.36557.010

As a test of the predictive power of our model, we designed a 250 bp-long sequence ('curved250') for which our model *a priori* predicted a high local curvature and strong plectoneme pinning (*Figure 3e*). When we subsequently synthesized and measured this construct, we indeed observed a pronounced peak in the plectoneme density (*Figure 3f*, blue). By contrast, when we constructed a 500 bp-long flat sequence without strongly curved regions ('flat500'), the model predicted no such peak, which again was verified experimentally (*Figure 3f*, black). These data demonstrate that the model can be used to identify potential plectoneme pinning sites in silico. Perhaps most strikingly, we found that a single highly curved DNA sequence of only 75 bp length was able to pin plectonemes (*Figure 3g*), consistent with the approximated tip loop size in our physical model (~73 bp). As a negative control, we did not observe any such pinning when we inserted a 75 bp-long flat DNA sequence (*Figure 3h*).

Finally, we wanted to verify that the intrinsic curvature, and not the GC/AT content, is the major determinant for plectoneme formation. Given that the earlier examples in *Figure 2f* clearly showed that some but not all AT-rich sequences can pin plectonemes, we designed some specifically GC-rich (i.e., AT-poor) sequences that should pin plectonemes. Because of the distribution of wedge angles available, GC-rich sequences tend to produce less intrinsic curvature over >10 bp sequences. To generate plectoneme pinning at a GC-rich sequence, we therefore inserted 8 repeats of a 75 bp-long GC-rich (~60%) insert in the middle of the flat500 sequence. As predicted by the model, the experimental data for this GC-rich curved sequence showed plectoneme pinning (*Figure 3i*), once more confirming that intrinsic curvature and not AT/GC content is the major determinant for plectoneme pinning.

## Transcription start sites localize plectonemes in prokaryotic genomes

Given the success of our physical model for predicting plectoneme localization, it is of interest to examine if the model identifies areas of high plectoneme density in genomic DNA that might directly relate to biological functions. Given that our model associates plectoneme pinning with high curvature, we were particularly interested to see what patterns might associate with specific genomic regions. For example, in prokaryotes, curved DNA has been observed to localize upstream of transcription start sites (TSS) (*Kanhere and Bansal, 2005*; *Olivares-Zavaleta et al., 2006*; *Pérez-Martín et al., 1994*). In eukaryotes, curvature is associated with the nucleosome positioning sequences found near promoters (*Tompitak et al., 2017*). However, given that our model requires highly curved DNA over long lengths of ~73 bp to induce plectoneme pinning, it was *a priori* unclear if the local curvature identified at promoter sites is sufficient to strongly influence the plectoneme density.

We first used the model to calculate the plectoneme density profile for the entire *E. coli* genome, revealing plectonemic hot spots spread throughout the genomic DNA (*Figure 4a*). Interestingly, we find that a substantial fraction of these hot spots are localized ~100 nucleotides upstream of all the transcription start sites (TSS) associated with confirmed genes in the RegulonDB database (*Figure 4b*, red) (*Gama-Castro et al., 2016*). We then performed a similar analysis of several other prokaryotic genomes (*Figure 4b*) (*Cortes et al., 2013*; *Irla et al., 2015*; *Papenfort et al., 2015*; *Zhou et al., 2015*). We consistently observe a peak upstream of the TSS, but the size of the peak varied substantially between species, indicating that different organisms rely on sequence-dependent plectoneme positioning to different extents. In one organism (*C. crescentus*), the signal was

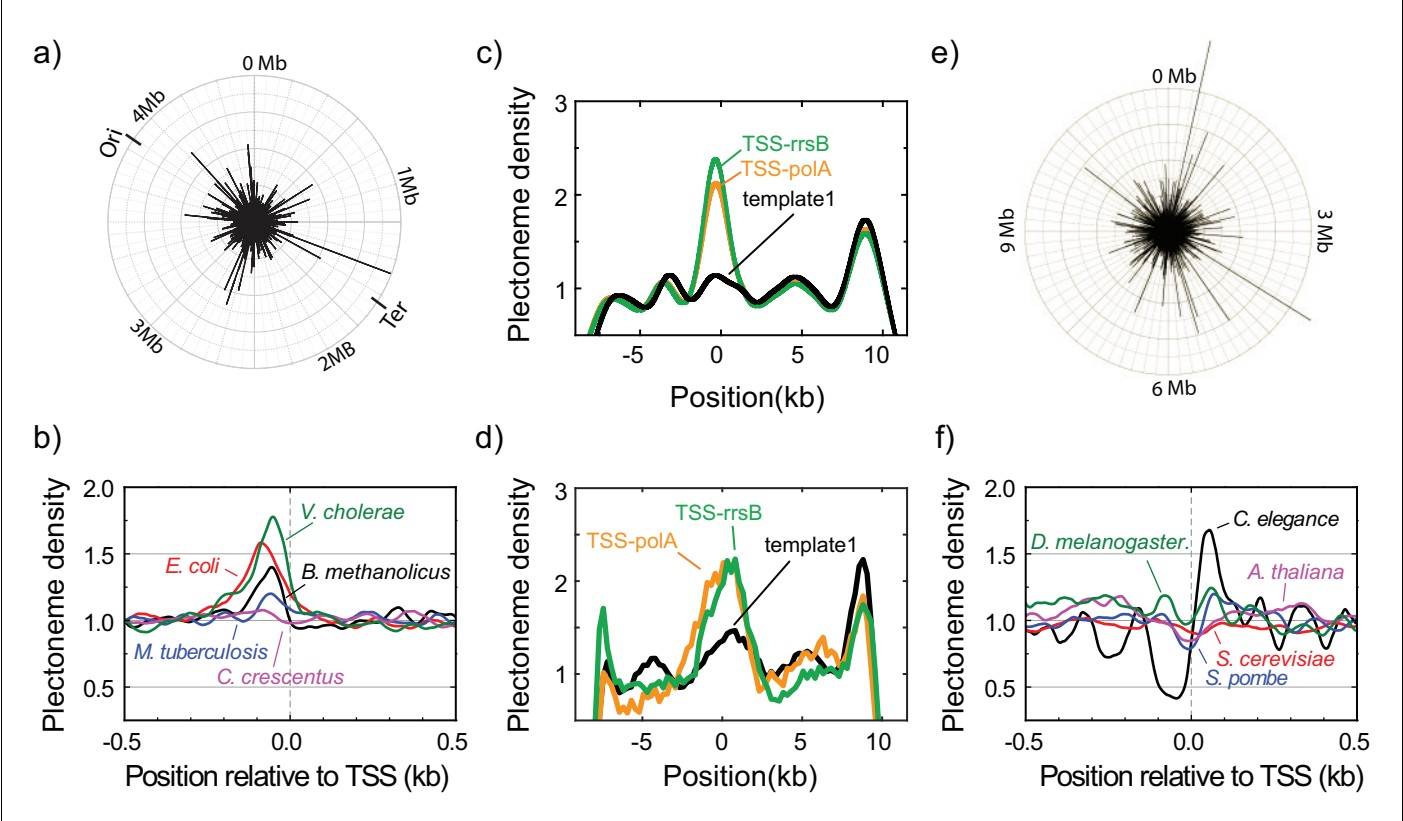

**Figure 4.** Plectonemes are enriched at prokaryotic transcription start sites. (a) The strength of plectoneme pinning calculated for the entire *E. coli* genome (4,639,221 bp; NC_000913). (b) Mean predicted plectoneme densities around transcription start sites (TSS) in prokaryotic genomes. The density profiles were smoothed over a 51 bp window. (c) Model-predicted and (d) experimentally measured plectoneme densities obtained for two selected TSS sites, TSS-rrsB and TSS-polA, which are E. coli transcription start sites encoding for 16S ribosomal RNA and DNA polymerase I, respectively. For comparison to experimental data, we smoothed the predicted plectoneme densities using a Gaussian filter (FWHM = 1600 bp) that approximates our spatial resolution. (n = 26, and 17 for TSS-rrsB, and TSS-polA, respectively) (e) Strength of plectoneme pinning calculated for the entire 12.1 Mb genome (i.e. all 16 chromosomes placed in sequential order) of *S. cerevisiae* (NC_001134). For quantitative comparison, we kept the radius of the outer circle the same as in (a). (f) Mean predicted plectoneme densities around the most representative TSS for each gene in several eukaryotic genomes. The density profiles are smoothed over a 51 bp window.

DOI: https://doi.org/10.7554/eLife.36557.011

too weak to detect at all. To experimentally confirm that these sequences represent plectonemic hot spots, we inserted two of these putative plectoneme-pinning sites from *E. coli* into template1. Gratifyingly, we indeed observed a strong pinning effect for these sequences in our single-molecule assay (*Figure 4c–d*).

Finally, we extended our analysis to eukaryotic organisms. Again we found plectonemic hotspots that were spread throughout the genome (*Figure 4e*). When averaging near the TSS (*Dreos et al., 2017*), we found a diverse range of plectoneme positioning signals (*Figure 4f*). While one organism (*S. cervisiae*) showed no detectable plectoneme positioning, most organisms showed both peaks and valleys indicating plectonemes were enriched but also depleted at different regions around the promoter. The features showed a weak periodicity consistent with the reported nucleosome repeat lengths (~150–260 bp) (*Jiang and Pugh, 2009*).

## Discussion

In this study, we reported direct experimental observations as well as a novel basic physical model for the sequence-structure relationship of supercoiled DNA. Our single-molecule ISD technique allowed a systematic analysis of sequences that strongly affect plectoneme formation. To explain the underlying mechanism, we developed a physical model that predicts the probability of plectoneme

pinning, based solely on the intrinsic curvature and the flexibility of a local region of the DNA. In the positive supercoiling regime (where no partial duplex melting is expected for the physiological range of tensions and torques), we identified the intrinsic curvature over a ~ 70 bp range as the primary factor that determines plectoneme pinning, while the flexibility alters the mechanics only minimally. Examining full genomes, we found that plectonemes are enriched at promoter sequences in *E. coli* and other prokaryotes, which suggests a role of genetically encoded supercoils in cellular function. Our findings reveal how a previously unrecognized 'hidden code' of intrinsic curvature governs the localization of local DNA supercoils, and hence the organization of the three-dimensional structure of the genome.

For a long time, researchers wondered whether DNA sequence may influence the plectonemic structure of supercoiled DNA. Structural and biochemical approaches identified special sequence patterns such as poly(A)-tracts that indicated plectoneme pinning (*Kremer et al., 1993*; *Laundon and Griffith, 1988*; *Pfannschmidt and Langowski, 1998*; *Tsen and Levene, 1997*). These early studies suggested that highly curved DNA can pin plectonemes, but the evidence was anecdotal and restricted to a handful of example sequences and it was not possible to establish a general rule for sequence-dependent plectoneme formation. Our high-throughput ISD assay, however, generated ample experimental data that enabled a comprehensive understanding of the underlying mechanism of the sequence-dependent plectoneme pinning.

Our physical modeling reveals that intrinsic curvature is the key structuring factor for determining the three-dimensional structure of supercoiled DNA. In contrast, although perhaps counter-intuitive, we found that the local flexibility is hardly relevant for plectoneme localization. Although highly flexible mismatched single-stranded regions have been shown to be able to act as a preferential position for plectoneme formation (*Dittmore et al., 2017*; *Ganji et al., 2016b*), the variations in the flexibility of duplex DNA due to sequence differences seem to produce very minor changes in the pinning probability.

Remarkably, although only the energy required to form the limited tip-loop region of ~73 bp is considered in our modeling, the model is capable of strikingly good qualitative predictions. In occasional cases, the model failed to reproduce the experimental results, giving some false negative predictions. A full statistical mechanical modeling of the plectonemic structures distributed across the DNA molecule should further improve the predictive power and accuracy, but will require significant computational resources and time.

Significant intrinsic curvatures are encoded in genomic DNA, as evident in our scans of both prokaryotic and eukaryotic genomes, which indicates its biological relevance. In support of this idea, an in silico study indeed suggested that curved prokaryotic promoters may control gene expression (*Gabrielian et al., 1999*). Moreover, early in vivo studies showed that curved DNA upstream to the promoter site affects gene expression levels (*Collis et al., 1989*; *McAllister and Achberger, 1989*). These in vivo studies suggested that curved DNA facilitates binding of RNA polymerase, an idea that is further supported by sharply bent DNA structures found around bound RNAP (*Rees et al., 1993*; *Tahirov et al., 2002*; *ten Heggeler and Wahli, 1985*; *Yin and Steitz, 2002*). In addition to this direct interaction of RNA polymerase and curved DNA, our results suggest an indirect effect, as the same curved DNA can easily pin a plectoneme that can further regulate the transcription initiation and elongation by structural re-arrangement of the promotor and coding regions.

Our analysis of prokaryotic genomes indicates that promoter sequences have evolved local regions with highly curved DNA that promote the localization of DNA plectonemes at these sites. There may be multiple reasons for this. For one, it may help to expose these DNA regions to the outer edge of the dense nucleoid, making them accessible to RNAP, transcription factors, and topoisomerases. Plectonemes may also play a role in the bursting dynamics of gene expression, since each RNAP alters the supercoiling density within a topological domain as it transcribes (*Chong et al., 2014*; *Kouzine et al., 2013*), adding or removing nearby plectonemes (*Liu and Wang, 1987*). In addition, by bringing distant regions of DNA close together, plectonemes may influence specific promoter-enhancer interactions to regulate gene expression (*Benedetti et al., 2014*). Finally, plectoneme tips may help RNA polymerase to initiate transcription, since the formation of an open complex also requires bending of the DNA (*ten Heggeler-Bordier et al., 1992*), a mechanism that was proposed as a universal method of regulating gene expression across all organisms (*Travers and Muskhelishvili, 2007*). The ability of our model to predict how mutations in the promoter sequence alter the plectoneme density opens up a new way to test these hypotheses.

Our analysis of eukaryotic genomes showed a greater diversity of behavior. The spacing of the peaks suggests that plectonemes may play a role in positioning nucleosomes, consistent with proposals that nucleosome positioning may rely on sequence-dependent signals near promoters (*Travers et al., 2010*). It is also broadly consistent with the universal topological model of plectoneme-RNAP interaction at promoters (*Travers and Muskhelishvili, 2007*), which proposes that the plectoneme tip forming upstream of the TSS in eukaryotes is positioned by nearby nucleosomes. The plectoneme signal encoded by intrinsic curvature could therefore indirectly position the promoter plectoneme tip by helping to organize these nearby nucleosomes.

In our study, we investigated the sequence-dependent behavior of plectonemes in a positively supercoiled state, although the technique can be extended to study negative supercoiling as well. For negative supercoils, plectoneme pinning can be influenced by both sequence-induced local curvature and local melting, which are hard to disentangle. Furthermore, although theoretical methods have been developed for the sequence dependence of the duplex stability of negatively supercoiled DNA (*Benham, 1990*; *Benham, 1992*), torsion-induced melting has been shown to exhibit complicated properties (*Vlijm et al., 2015*). The model that we have developed for positive supercoils should not be very sensitive to the handedness of supercoiling, since the dinucleotide curvature parameters are not strongly perturbed at these torques. We therefore expect the model to also capture curvature-dependent effects on pinning of negative plectonemes too.

The above findings demonstrate that DNA contains a previously hidden 'code' that determines the local intrinsic curvature and consequently governs the locations of plectonemes. These plectonemes can organize DNA within topological domains, providing fine-scale control of the three-dimensional structure of the genome (*Le et al., 2013*). The model and assay described here make it possible both to predict how changes to the DNA sequence will alter the distribution of plectonemes and to investigate the DNA supercoiling behavior at specific sequences empirically. Using these tools, it will be interesting to explore how changes in this plectoneme code affect levels of gene expression and other vital cellular processes.

## Materials and methods

### Preparation of DNA molecules of different sequences

Full sequences of all DNA molecules are given in *Supplementary file 2*. All DNA molecules except 'template 2' in *Figure 1* were prepared by ligating four or five DNA fragments, respectively: 1) 'Cy5-biotin handle', 2) '8.4 kb fragment', [3] 'Sequence of Interest',] 4) '11.2 kb fragment', and 5) 'biotin handle' (*Figure 1—figure supplement 1b*). The 'Cy5-biotin handle' and 'biotin handle' were prepared by PCR methods in the presence of Cy5-modified and/or biotinylated dUTP (aminoallyl-dUTP-Cy5 and biotin-16-dUTP, Jena Bioscience). The '8kb-fragment' and '11 kb fragment' were prepared by PCR on Unmethylated Lambda DNA (Promega). These fragments were cloned into pCR-XL using the TOPO XL PCR cloning kit (Invitrogen) generating pCR-XL-11.2 and pCR-XL-8.4 (*Ganji et al., 2016b*). The fragments were PCR amplified and then digested with BsaI restriction enzyme, respectively (*Supplementary file 3*). The 'Sequence of Interest' was made by PCR on different templates. Template two in *Figure 1C*-black and 1e was made from a digested fragment of an engineered plasmid pSuperCos-λ1,2 with XhoI and NotI-HF (*van Loenhout et al., 2012*). The digested fragment was further ligated with biotinylated PCR fragments on XhoI side and a biotinylated-Cy5 PCR fragment on the NotI-HF (*Supplementary file 4*). All the DNA samples were gel-purified before use.

### Dual-color epifluorescence microscopy

Details of our experimental setup are described previously (*Ganji et al., 2016a*; *Ganji et al., 2016b*). Briefly, a custom-made epifluorescence microscopy equipped with two lasers (532 nm, Samba, Cobolt and 640 nm, MLD, Cobolt) and an EMCCD camera (Ixon 897, Andor) is used to image fluorescently labeled DNA molecules. For the wide-field, epifluorescence-mode illumination on the sample surface, the two laser beams were collimated and focused at the back-focal plane of an objective lens (60x UPLSAPO, NA 1.2, water immersion, Olympus). Back scattered laser light was filtered by using a dichroic mirror (Di01-R405/488/543/635, Semrock) and the fluorescence signal was spectrally separated by a dichroic mirror (FF635-Di02, Semrock) for the SxO channel and Cy5 channel. Two band-pass filters (FF01-731/137, Semrock, for SxO) and FF01-571/72, Semrock, for Cy5) were

employed at each fluorescence channel for further spectral filtering. Finally, the fluorescence was imaged on the CCD camera by using a tube lens (f = 200 mm). All the measurements were performed at room temperature.

## Intercalation-induced supercoiling of DNA (ISD)

A quartz slide and a coverslip were coated with polyethlyleneglycol (PEG) to suppress nonspecific binding of DNA and SxO. 2% of the PEG molecules were biotinylated for the DNA immobilization. The quartz slide and coverslip were sandwiched with a double-sided tape such that a 100 μm gap between the slide and coverslip forms a shallow sample chamber with flow control. Two holes serving as the inlet and outlet of the flow were placed on the slide glass. Typically, a sample chamber holds 10 μl of solution.

Before DNA immobilization, we incubated the biotinylated PEG surface with 0.1 mg/ml streptavidin for 1 min. After washing unbound streptavidin by flowing 100 μl of buffer A (40 mM TrisHCl pH 8.0, 20 mM NaCl, and 0.2 mM EDTA), we flowed the end-biotinylated DNA diluted in buffer A into the sample chamber at a flow rate of 50 μl/min. The concentration of the DNA (typically ~10 pM) was empirically chosen to have an optimal surface density for single DNA observation. Immediately after the flow, we further flowed 200 μl of buffer A at the same flow rate, resulting in stretched, doubly tethered DNA molecules (*Figure 1a* and *Figure 1—figure supplement 1a*) of which end-to-end extension can be adjusted by the flow rate. We obtained the DNA lengths of around 60–70% of its contour length (*Figure 1—figure supplement 2a*), which corresponds to a force range of 2–4 pN (*Ganji et al., 2016b*). We noted that SxO does not exhibit any sequence preference when binding to relaxed DNA, allowing us to back out the amount of DNA localized within a diffraction-limited spot from the total fluorescence intensity.

After immobilization of DNA, we flowed in 30 nM SxO (S11368, Thermo Fisher) in an imaging buffer consisting of 40 mM Tris-HCl, pH 8.0, 20 mM NaCl, 0.4 mM EDTA, 2 mM trolox, 40 μg/ml glucose oxidase, 17 μg/ml catalase, and 5% (w/v) D-dextrose. Fluorescence images were taken at 100 ms exposure time for each frame. The 640 nm laser was used for illuminated for the first 10 frames (for Cy5 localization), followed by continuous 532 nm laser illumination afterwards. From our previous study, we noted that SxO locally unwinds DNA and extends the contour length (*Figure 1—figure supplement 1a*), but does not otherwise affect the mechanical properties of the DNA (*Ganji et al., 2016b*). Based on the same previous work and assuming that each intercalating dye reduces the twist at the local dinucleotide to zero, we estimate that roughly 1 SxO is bound on every 26 base-pairs of DNA. We note that the numbers of plectoneme nucleation and termination events along supercoiled DNA were equal (*Figure 1—figure supplement 2b*), which is characteristic of a system at equilibrium. Furthermore, we verified that increasing the NaCl concentration from 20 mM to 150 mM NaCl did not result in any significant difference in the observed plectoneme density results, indicating that the plectoneme density is not dependent on the ionic strength (*Figure 2—figure supplement 1f*).

## Data analysis

Analysis of the data was carried out using custom-written Matlab routines, as explained in our previous report (*Ganji et al., 2016b*). Briefly, we averaged the first ten fluorescence images to determine the end positions of individual DNA molecules. We identify the direction of the DNA molecules by 640 nm illumination at the same field of view, which identifies the Cy5-labelled DNA end. Then, the fluorescence intensity of the DNA at each position along the length was determined by summing up 11 neighboring pixels perpendicular to the DNA at that position. The median value of the pixels surrounding the molecule was used to correct the background of the image. The resultant DNA intensity was normalized to the total intensity sum of the DNA for each frame to compensate for photobleaching of SxO. We recorded more than 300 frames, each taken with a 100 msec exposure time, and built an intensity kymograph by aligning the normalized intensity profiles in time. Supercoiled DNA intensity profiles, that is single lines in the intensity kymograph, were converted to DNA-density profiles by comparing the intensity profile of supercoiled DNA to that of the corresponding relaxed DNA. Specifically, the ratio between the cumulative intensities of all the pixels in the right and the left-hand sides of each position of the DNA was first determined. To find the genomic position (i.e. base pair position) of the peak, we compared this ratio with that obtained after torsional

relaxation of the molecule of which the pixel position is the same with the genomic position under the given constant tension (*Ganji et al., 2016b*). The torsionally relaxed intensity profile was obtained after the plectoneme measurements by increasing the excitation laser power that yielded a photo-induced nick of the DNA.

The position of a plectoneme is identified by applying a threshold algorithm to the DNA density profile. A median of the entire DNA density kymograph was used as the background DNA density. The threshold for plectoneme detection was set at 25% above the background DNA density. Peaks that sustain at least three consecutive time frames (i.e., ≥300 ms) were selected as plectonemes. After identifying all the plectonemes, the probability of finding a plectoneme at each position (250 bp-long segment) along the DNA in base-pair space was calculated by counting the total number of plectonemes at each position (segment) divided by the total observation time. The probability density is then further normalized to its mean value across the DNA molecule to build a plectoneme density. Note that the plectoneme density represents the relative propensity of plectoneme formation at different regions within a DNA molecule, which is insensitive to the length of the DNA as well as the linking number. Typically, more than 20 DNA molecules were measured for each DNA sample and the averaged plectoneme densities were calculated with a weight given by the observation time of each molecule. The analysis code written in Matlab (The MathWorks, Inc.) is freely available from GitHub (*Kim, 2018*; copy archived at https://github.com/elifesciences-publications/Plectoneme_analysis).

## Plectoneme tip-loop size estimation and bending energetics

An important component of our model is to determine the energy involved in bending the DNA at the plectoneme tip. We first estimate the mean size of a plectoneme tip-loop from the energy stored in an elastic polymer with the same bulk features of DNA. For the simplest case, we first consider a circular loop (360°) formed in DNA under tension. The work associated with shortening the end-to-end length of DNA to accommodate the loop is

$$W = rFN$$

where $F$ is the tension across the polymer, $r$ is the base pair rise (0.334 nm for dsDNA), and $N$ is the number of base pairs. The bending energy is

$$E_{bend} = \frac{2\pi^2 k_B T A}{rN}$$

where $k_B$ is the Boltzmann constant, $A$ is the bulk persistence length (50 nm for dsDNA). Hence, we obtain an expression for the total energy:

$$E_{total} = rFN + \frac{2\pi^2 k_B T A}{rN} = k_B T (CN + B_{360}/N)$$

Taking the derivative of $E_{total}$ with respect to $N$ and setting it to zero gives the formula:

$$N = \sqrt{\frac{B_{360}}{C}}$$

Here, the values of the constants are:

$$C = \frac{F}{12.16 pN}$$

$$B_{360} = 2955$$

So, at 3 pN we get:

$$N = \sqrt{\frac{B_{360}}{C}} = 109$$

If the loop at the end of the plectoneme is held at the same length but only bent to form a partial circle, the work needed to accommodate the loop will remain the same but the bending energy will

be lower, scaling quadratically with the overall bend angle. For a plectoneme tip, a 240° loop is sufficient to match the angle of the DNA in the stem of the plectoneme. The preferred length of a 240° loop is therefore:

$$N = \sqrt{\frac{B_{240}}{C}} = 73$$

where:

$$B_{240} = B_{360} \left(\frac{240°}{360°}\right)^2$$

## Physical model predicting the plectoneme density

A full model must explicitly account for the fact that DNA is not a homogeneous polymer. Instead, each DNA sequence has (1) intrinsic curvature and (2) a variable flexibility. Both 1 and 2 depend on the dinucleotide sequences at each location. Note also that we can bend the DNA along any vector normal to the path of the DNA, which describes a circle spanning the full 360° surrounding the DNA strand. We must therefore specify the direction of bending $\phi$ when calculating the bend energy, and we define $\phi = \phi_B$ to be the bend direction that aligns with the intrinsic curvature.

The intrinsic curvature can be estimated from the dinucleotide content of the DNA (*Figure 3a*). Several studies have attempted to measure the optimal set of dinucleotide parameters (i.e. tilt, roll, and twist) that most closely predict actual DNA conformations (*Balasubramanian et al., 2009*; *Bolshoy et al., 1991*; *Morozov et al., 2009*; *Olson et al., 1998*). We find that the parameter set by Balasubramanian et al., produces the closest match to our experimental data when plugged into our model (*Balasubramanian et al., 2009*). Using these parameters (see *Supplementary file 1*), we first calculate the winding ground state path traced out by the entire DNA strand. We then determine the intrinsic curvature, $\theta(N,i)$, across a given stretch of N nucleotides centered at position *i* on the DNA by comparing tangent vectors at the start and end of that stretch. Tangent vectors are calculated over an 11 bp window (one helical turn,~3.7 nm). Note that the intrinsic curvature, defined by $\theta(N,i)$, also determines the preferred bend direction $\phi_B$.

The flexibility of the DNA also varies with position. The flexibility of the tilt and roll angles between neighboring dinucleotide has been estimated by MD simulations (*Lankas et al., 2003*). Using these numbers, we can add the roll-tilt covariance matrices for a series of nucleotides (each rotated by the twist angle) to calculate the local flexibility of a given stretch of DNA. The flexibility also depends on the direction of bending. The summed covariance matrix allows us to estimate a local persistence length $A(N,l,\phi)$.

By combining the local bend angle $\theta(N,i)$ and the local persistence length $A(N,l,\phi)$, we are now able to calculate the energy needed to bend a given stretch of DNA to 240°. When the DNA is bent in the preferred curvature direction, this bending energy becomes:

$$\frac{E_{bend}(N,i,\phi_B)}{K_B T} = \left(\frac{2}{3}\right)^2 \frac{2\pi^2 A(N,i,\phi_B)}{0.334nm \cdot N} \left(1 - \frac{\theta(N,i)}{240°}\right)^2$$

More generally, we can bend the DNA in any direction, in which case the bending energy can be calculated using the law of cosines:

$$\frac{E_{bend}(N,i,\phi)}{K_B T} = \left(\frac{2}{3}\right)^2 \frac{2\pi^2 A(N,i,\phi)}{0.334nm \cdot N} \left[1 + \left(\frac{\theta(N,i)}{240°}\right)^2 - 2\left(\frac{\theta(N,i)}{240°}\right) cos(\phi - \phi_B)\right]$$

The first formula is the special case when $\phi = \phi_B$.

Because both $A(N,i, \phi)$ and $\theta(N,i)$ are sequence dependent, the loop size and bend direction that minimizes the free energy will also be sequence dependent. Rather than trying to find the parameters that give a maximum likelihood at each position along the template, we find that it is more efficient to calculate the relative probabilities of loops spanning a range of sizes and bend directions. We first calculate the energy associated with each loop using:

$$\frac{E_{total}(N,i,\phi)}{k_BT} = \frac{rF}{k_BT}N + \frac{E_{bend}(N,i,\phi)}{k_BT}$$

We then assign each of these bending conformations a Boltzmann weight:

$$W(N,i,\phi) = exp\left(-\frac{E_{total}(N,i,\phi)}{k_BT}\right)$$

Finally, we sum over all the different bending conformations to get the total weight assigned to the formation of a plectoneme at a specific location $i$ on the template:

$$W_{tot}(i) = \sum_{N,\phi} W(N,i,\phi)$$

Because the direction $\phi$ is a continuous variable and the length of the loop can range strongly, there are a very large number of bending conformations to sum over. However, because of the exponential dependence on energy, only conformations near the maximum likelihood value in phase space will contribute significantly to the sum. For an isotropic DNA molecule, the maximum likelihood should occur at $N = 73$ and $\phi = \phi_B$. We therefore sum over parameter values that span this point in phase space. Our final model sums over eight bending directions (i.e. at every 45°, starting from $\phi = \phi_B$) and calculates loop sizes over a range from 40 bp to 120 bp at 8 bp increments. We verified that the predictions of the model were stable if we increased the range of the loop sizes considered or increased the density of points sampled in phase space, implying that the range of values used was sufficient.

For a fair comparison to experimental data, all predicted plectoneme densities that are presented were smoothened using a Gaussian filter (FWHM = 1600 bp) that approximates our spatial resolution. The code for the model prediction is freely available from GitHub (*Abbondanzieri, 2018*; copy archived at https://github.com/elifesciences-publications/Plectoneme_prediction).

## Acknowledgments

The data reported in the paper are available from the corresponding authors upon request. We acknowledge valuable discussions with Helmut Schiessel and Ard Louis. We thank Jacob Kerssemakers for helpful discussion and data analysis codes. This work was supported by the ERC Advanced Grant SynDiv [grant number 669598 to CD]; the Netherlands Organization for Scientific Research (NWO/OCW) [as part of the Frontiers of Nanoscience program], and the ERC Marie Curie Career Integration Grant [grant number 304284 to EA].

## Additional information

### Funding

| Funder | Grant reference number | Author |
|---|---|---|
| H2020 European Research Council | 669598 | Cees Dekker |
| The Netherlands Organization for Scientific Research | The Frontiers of Nanoscience program | Elio Abbondanzieri |
| H2020 European Research Council | 304284 | Elio Abbondanzieri |

The funders had no role in study design, data collection and interpretation, or the decision to submit the work for publication.

### Author contributions

Sung Hyun Kim, Mahipal Ganji, Conceptualization, Data curation, Software, Formal analysis, Validation, Investigation, Visualization, Methodology, Writing—original draft, Writing—review and editing; Eugene Kim, Data curation, Formal analysis; Jaco van der Torre, Resources, Investigation, Writing—

original draft; Elio Abbondanzieri, Conceptualization, Data curation, Software, Formal analysis, Supervision, Funding acquisition, Validation, Investigation, Visualization, Methodology, Writing— review and editing; Cees Dekker, Conceptualization, Supervision, Funding acquisition, Validation, Investigation, Project administration, Writing—review and editing

### Author ORCIDs
Sung Hyun Kim http://orcid.org/0000-0001-9272-7036
Mahipal Ganji http://orcid.org/0000-0001-8176-3322
Cees Dekker http://orcid.org/0000-0001-6273-071X

### Decision letter and Author response
Decision letter https://doi.org/10.7554/eLife.36557.054
Author response https://doi.org/10.7554/eLife.36557.055

## Additional files

### Supplementary files
• Supplementary file 1. Dinucleotide parameters used in the physical model.
DOI: https://doi.org/10.7554/eLife.36557.012
• Supplementary file 2. DNA sequences of the inserts
DOI: https://doi.org/10.7554/eLife.36557.013
• Supplementary file 3. Materials used for DNA template one with and without inserts
DOI: https://doi.org/10.7554/eLife.36557.014
• Supplementary file 4. Materials used for DNA template 2
DOI: https://doi.org/10.7554/eLife.36557.015
• Transparent reporting form
DOI: https://doi.org/10.7554/eLife.36557.016

### Data availability
All data generated or analysed during this study are included in the manuscript and supporting files. The previously published genome data for E. coli used in Figure 4B can be accessed here http://regulondb.ccg.unam.mx/menu/download/datasets/files/PromoterSet.txt; V. cholerae here http://www.pnas.org/highwire/filestream/618514/field_highwire_adjunct_files/2/pnas.1500203112.sd02.xlsx; B. methanolicus here https://www.ncbi.nlm.nih.gov/pmc/articles/PMC4342826/bin/12864_2015_1239_MOESM2_ESM.xlsx; M. tuberculosis here https://ars.els-cdn.com/content/image/1-s2.0-S2211124713006153-mmc2.xlsx; and C. crescentus here https://doi.org/10.1371/journal.pgen.1004831.s012. The previously published genome data for D. melanogaster, C. elegans, A. thaliana, S. cerevisiae, and S. pombe used in Figure 4E can be accessed using the Eukaryotic Promotor Database (https://epd.vital-it.ch).

The following previously published datasets were used:

| Author(s) | Year | Dataset title | Dataset URL | Database and Identifier |
|---|---|---|---|---|
| Riley M, Abe T, Arnaud MB, Berlyn MK, Blattner FR, Chaudhuri RR, Glasner JD, Horiuchi T, Keseler IM, Kosuge T, Mori H, Perna NT, Plunkett G III, Rudd KE, Serres MH, Thomas GH, Thomson NR, Wishart D, Wanner BL | 2006 | Whole genome sequence data: E. Coli | https://www.ncbi.nlm.nih.gov/genome/?term=NC_000913.3 | NCBI Genome, NC_000913.3 |
| Goffeau A, Barrell BG, Bussey H, Da- | 1996 | Whole genome sequence data: S. cerevisiae (Chr I) | https://www.ncbi.nlm.nih.gov/nuccore/NC_ | NCBI Nucleotide, NC_001133.9 |

| | | | | |
|---|---|---|---|---|
| vis RW, Dujon B, Feldmann H, Galibert F, Hoheisel JD, Jacq C, Johnston M, Louis EJ, Mewes HW, Murakami Y, Philippsen P, Tettelin H, Oliver SG | | | 001133.9 | |
| Goffeau A, Barrell BG, Bussey H, Davis RW, Dujon B, Feldmann H, Galibert F, Hoheisel JD, Jacq C, Johnston M, Louis EJ, Mewes HW, Murakami Y, Philippsen P, Tettelin H, Oliver SG | 1996 | Whole genome sequence data: S. cerevisiase (Chr II) | https://www.ncbi.nlm.nih.gov/nuccore/NC_001134.8 | NCBI Nucleotide, NC_001134.8 |
| Goffeau A, Barrell BG, Bussey H, Davis RW, Dujon B, Feldmann H, Galibert F, Hoheisel JD, Jacq C, Johnston M, Louis EJ, Mewes HW, Murakami Y, Philippsen P, Tettelin H, Oliver SG | 1996 | Whole genome sequence data: S. cerevisiase (Chr III) | https://www.ncbi.nlm.nih.gov/nuccore/NC_001135.5 | NCBI Nucleotide, NC_001135.5 |
| Jacq C, Alt-Mörbe J, Andre B, Arnold W, Bahr A, Ballesta JP, Bargues M, Baron L, Becker A, Biteau N, Blöcker H, Blugeon C, Boskovic J, Brandt P, Brückner M, Buitrago MJ, Coster F, Delaveau T, del Rey F, Dujon B, Eide LG, Garcia-Cantalejo JM, Goffeau A | 1997 | Whole genome sequence data: S. cerevisiase (Chr IV) | https://www.ncbi.nlm.nih.gov/nuccore/NC_001136.10 | NCBI Nucleotide, NC_001136.10 |
| Dietrich FS, Mulligan J, Hennessy K, Yelton MA, Allen E, Araujo R, Aviles E, Berno A, Brennan T, Carpenter J, Chen E, Cherry JM, Chung E, Duncan M, Guzman E, Hartzell G, Hunicke-Smith S, Hyman RW, Kayser A, Komp C, Lashkari D, Lew H, Lin D, Mosedale D, Davis RW | 1997 | Whole genome sequence data: S. cerevisiase (Chr V) | https://www.ncbi.nlm.nih.gov/nuccore/NC_001137.3 | NCBI Nucleotide, NC_001137.3 |
| Goffeau A, Barrell BG, Bussey H, Davis RW, Dujon B, Feldmann H, Galibert F, Hoheisel JD, Jacq C, Johnston M, Louis EJ, Mewes HW, Murakami Y, Philippsen | 1996 | Whole genome sequence data: S. cerevisiase (Chr VI) | https://www.ncbi.nlm.nih.gov/nuccore/NC_001138.5 | NCBI Nucleotide, NC_001138.5 |

| | | | | |
|---|---|---|---|---|
| P, Tettelin H, Oliver SG | | | | |
| Tettelin H, Agostoni Carbone ML, Albermann K, Albers M, Arroyo J, Backes U, Barreiros T, Bertani I, Bjourson AJ, Brückner M, Bruschi CV, Carignani G, Castagnoli L, Cerdan E, Clemente ML, Coblenz A, Coglievina M, Coissac E, Defoor E, Del Bino S, Delius H, Delneri D, de Wergifosse P, Dujon B, Kleine K | 1997 | Whole genome sequence data: S. cerevisiase (Chr VII) | https://www.ncbi.nlm. nih.gov/nuccore/NC_ 001139.9 | NCBI Nucleotide, NC_001139.9 |
| Goffeau A, Barrell BG, Bussey H, Davis RW, Dujon B, Feldmann H, Galibert F, Hoheisel JD, Jacq C, Johnston M, Louis EJ, Mewes HW, Murakami Y, Philippsen P, Tettelin H, Oliver SG | 1996 | Whole genome sequence data: S. cerevisiase (Chr VIII) | https://www.ncbi.nlm. nih.gov/nuccore/NC_ 001140.6 | NCBI Nucleotide, NC_001140.6 |
| Churcher C, Bowman S, Badcock K, Bankier A, Brown D, Chillingworth T, Connor R, Devlin K, Gentles S, Hamlin N, Harris D, Horsnell T, Hunt S, Jagels K, Jones M, Lye G, Moule S, Odell C, Pearson D, Rajandream M, Rice P, Rowley N, Skelton J, Smith V, Barrell B | 1997 | Whole genome sequence data: S. cerevisiase (Chr IX) | https://www.ncbi.nlm. nih.gov/nuccore/NC_ 001141.2 | NCBI Nucleotide, NC_001141.2 |
| Goffeau A, Barrell BG, Bussey H, Davis RW, Dujon B, Feldmann H, Galibert F, Hoheisel JD, Jacq C, Johnston M, Louis EJ, Mewes HW, Murakami Y, Philippsen P, Tettelin H, Oliver SG | 1996 | Whole genome sequence data: S. cerevisiase (Chr X) | https://www.ncbi.nlm. nih.gov/nuccore/NC_ 001142.9 | NCBI Nucleotide, NC_001142.9 |
| Goffeau A, Barrell BG, Bussey H, Davis RW, Dujon B, Feldmann H, Galibert F, Hoheisel JD, Jacq C, Johnston M, Louis EJ, Mewes HW, Murakami Y, Philippsen P, Tettelin H, Oliver SG | 1996 | Whole genome sequence data: S. cerevisiase (Chr XI) | https://www.ncbi.nlm. nih.gov/nuccore/NC_ 001143.9 | NCBI Nucleotide, NC_001143.9 |
| Johnston M, Hillier L, Riles L, Albermann K, André B, Ansorge W, Benes | 1997 | Whole genome sequence data: S. cerevisiase (Chr XII) | https://www.ncbi.nlm. nih.gov/nuccore/NC_ 001144.5 | NCBI Nucleotide, NC_001144.5 |

V, Brückner M, Delius H, Dubois E, Düsterhöft A, Entian KD, Floeth M, Goffeau A, Hebling U, Heumann K, Heuss-Neitzel D, Hilbert H, Hilger F, Kleine K, Kötter P, Louis EJ, Messenguy F, Mewes HW, Hoheisel JD

| | | | | |
|---|---|---|---|---|
| Bowman S, Churcher C, Badcock K, Brown D, Chillingworth T, Connor R, Dedman K, Devlin K, Gentles S, Hamlin N, Hunt S, Jagels K, Lye G, Moule S, Odell C, Pearson D, Rajandream M, Rice P, Skelton J, Walsh S, Whitehead S, Barrell B | 1997 | Whole genome sequence data: S. cerevisiase (Chr XIII) | https://www.ncbi.nlm.nih.gov/nuccore/NC_001145.3 | NCBI Nucleotide, NC_001145.3 |
| Philippsen P, Kleine K, Pöhlmann R, Düsterhöft A, Hamberg K, Hegemann JH, Obermaier B, Urrestarazu LA, Aert R, Albermann K, Altmann R, André B, Baladron V, Ballesta JP, Bécam AM, Beinhauer J, Boskovic J, Buitrago MJ, Bussereau F, Coster F, Crouzet M, D'Angelo M, Dal Pero F, De Antoni A, Del Rey F, Doignon F, Domdey H, Dubois E, Fiedler T, Fleig U, Floeth M, Fritz C, Gaillardin C, Garcia-Cantalejo JM, Glansdorff NN, Goffeau A, Gueldener U, Herbert C, Heumann K, Heuss-Neitzel D, Hilbert H, Hinni K, Iraqui Houssaini I, Jacquet M, Jimenez A, Jonniaux JL, Karpfinger L, Lanfranchi G, Lepingle A, Levesque H, Lyck R, Maftahi M, Mallet L, Maurer KC, Messenguy F, Mewes HW, Mösti D, Nasr F, Nicaud JM, Niedenthal RK, Pandolfo D, Piérard A, Piravandi E, Planta RJ, Pohl TM, Purnelle B, Rebischung C, Remacha M, Revuelta JL, | 1997 | Whole genome sequence data: S. cerevisiase (Chr XIV) | https://www.ncbi.nlm.nih.gov/nuccore/NC_001146.8 | NCBI Nucleotide, NC_001146.8 |

| Rinke M, Saiz JE, Sartorello F, Scherens B, Sen-Gupta M, Soler-Mira A, Urbanus JH, Valle G, Van Dyck L, Verhasselt P, Vierendeels F, Vissers S, Voet M, Volckaert G, Wach A, Wambutt R, Wedler H, Zollner A, Hani J | | | | |
| Dujon B, Albermann K, Aldea M, Alexandraki D, Ansorge W, Arino J, Benes V, Bohn C, Bolotin-Fukuhara M, Bordonné R, Boyer J, Camasses A, Casamayor A, Casas C, Chéret G, Cziepluch C, Daignan-Fornier B, Dang DV, de Haan M, Delius H, Durand P, Fairhead C, Feldmann H, Gaillon L, Kleine K | 1997 | Whole genome sequence data: S. cerevisiase (Chr XV) | https://www.ncbi.nlm.nih.gov/nuccore/NC_001147.6 | NCBI Nucleotide, NC_001147.6 |
| Bussey H, Storms RK, Ahmed A, Albermann K, Allen E, Ansorge W, Araujo R, Aparicio A, Barrell B, Badcock K, Benes V, Botstein D, Bowman S, Brückner M, Carpenter J, Cherry JM, Chung E, Churcher C, Coster F, Davis K, Davis RW, Dietrich FS, Delius H, DiPaolo T, Hani J | 1997 | Whole genome sequence data: S. cerevisiase (Chr XVI) | https://www.ncbi.nlm.nih.gov/nuccore/NC_001148.4 | NCBI Nucleotide, NC_001148.4 |
| Foury F, Roganti T, Lecrenier N, Purnelle B | 1998 | Whole genome sequence data: S. cerevisiase | https://www.ncbi.nlm.nih.gov/nuccore/NC_001224.1 | NCBI Nucleotide, NC_001224.1 |

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
