## [Decision Letter]

Thank you for submitting your article "DNA sequence encodes the position of DNA supercoils" for consideration by *eLife*. Your article has been reviewed by three peer reviewers, one of whom is a member of our Board of Reviewing Editors, and the evaluation has been overseen by Naama Barkai as the Senior Editor. The reviewers have opted to remain anonymous.

The reviewers have discussed the reviews with one another and the Reviewing Editor has drafted this decision to help you prepare a revised submission.

The work here by Kim et al. describes a technically impressive approach to identify the relationship between DNA sequence and the positioning of DNA supercoils. In agreement with many earlier studies, the authors suggest that plectonemes are positioned by and formed at intrinsically curved DNA sequences. The authors additionally suggest that AT content and sequence flexibility contribute minimally, if at all, to supercoil positioning. Lastly, the authors develop a model and calculate plectoneme density across several genomes. Using this approach, they identify a correlation between curved sequences and transcriptional start site (TSS) location in several organisms and show that two of these sequences can indeed position supercoils in vitro. The assay and many of the underlying concepts have been previously reported, but the systematic study of diverse sequences and direct comparisons to a detailed model based on intrinsic curvature are, to our knowledge, novel. The work complements other recent studies that have explored pinning due to mismatches, general theory, and measurements based on overall DNA extension rather than direct imaging of plectonemes.

Although the reviewers were generally enthusiastic, there were some issues that arose during review and discussion that should be addressed in a revision. The first of these will require additional experiments. The others could, but in some cases may also be dealt with through changes in the text and presentation of the data:

1) While this work shows that AT content is not required for plectoneme formation, the authors' conclusion that the curvature is the predominant driver of supercoil formation is based on a limited number of sequence manipulations, with all insertions tested in a single longer context (template 1) and only one synthetic designed sequence tested after developing the model. A deeper analysis (changing the AT content, length, amount of curvature or eliminating curvature entirely) of their template is important to understand the sequence properties that are necessary for plectoneme formation. Since short 60-90 bp segments are sufficient for supercoil pinning (Kremer et al., 1993) and the authors' predict that a 73 bp region should be sufficient to induce pinning, shorter segments than the ones presented (250-450 bp Figure 3E, 4C) should be directly tested. Working with a GC-rich template, in particular, would improve the generality of the results.

2) Related to the point above, the authors state that "the model qualitatively represented the experimental data for all sequences that were tested", but it appears this comparison is only shown for a limited subset of the sequences. Given that the paper is centrally focused on proposing a general model, it would be appropriate to show comparisons with all sequences, to facilitate evaluation of the model's successes and limitations. In particular, it does not seem that model predictions are shown for about a dozen of the sequence variants for which experiments are reported:

– template 2

– template 1 + seqA, seqB, seqC;

– seqB with base shuffling and with AT-tract shuffling (the latter in particular is a central experiment for the paper)

– template 1 + variable length insertions (.25, 1, 2, 3, 3.9 kb)

3) The Introduction of the manuscript is somewhat misleading. An existing model in the field is that curved sequences induce supercoil formation (many of these papers are cited in the manuscript, see Kremer et al., 1993, Laundon and Griffith, 1988, Pfannschmidt and Langowski, 1998, Tsen and Levene, 1997, and Pavlicek et al., 2004, among others), however the authors do not acknowledge this. The authors are correct that a major shortcoming of these earlier works is their use of phased AT-tracts in generating a curved sequence, however, the introduction should be written to clarify this for the reader. Some of the references are also misattributed. In the second paragraph of the Introduction, Eslami-Mossallam et al., 2016, Pasi and Lavery, 2016, and Wang et al., 2017, are computational studies not biochemical and structural studies, while Kremer et al., 1993, Laundon and Griffith, 1988, Pfannschmidt and Langowski, 1998, and Tsen and Levene, 1997, used biochemical and structural approaches in addition to in silico modeling.

4) How the plectoneme density (e.g., Figure 1D) is calculated was buried in the Materials and methods and the prior paper from this group, making it a little difficult to understand what was done. Given the centrality of plectoneme density plots, it would be useful for the definition, calculation procedure, and motivation for choosing this metric to be explained clearly up front. The Materials and methods mention the calculation of the size of each plectoneme (how much writhe is constrained in the molecule at each position) but it is unclear whether the plectoneme size is ever used in any of the reported analysis. Relatedly, how do AT-content, curvature, and other sequence features affect plectoneme formation/termination kinetics and the amount of writhe per plectoneme? Finally, can the authors quantify sequence effects on plectoneme "slithering" (diffusion)? They mention that the model predicts the expected degree of roughness to explain plectoneme diffusion rates, but do not comment on whether sequence differences cause predictable changes to these rates.

5) The experiments here are performed in the presence of positive supercoiling. There could be important differences under negative supercoiling, relevant to biological contexts. For example, the authors present evidence for intrinsic curvature as a dominant contributor to sequence-dependent plectonemic pinning. An alternative proposal (Matek et al.) is that duplex stability plays an important role, via the formation of plectoneme tip bubbles. If these occur, they are expected to occur more readily in negatively supercoiled DNA, which favors strand separation. Ideally, the authors would provide any evidence addressing whether plectoneme pinning occurs at the same locations/with the same sequence determinants for positively and negatively supercoiled DNA. Clearly the high-throughput assay used here can only interrogate positively supercoiled DNA, so this would require a lower-throughput assay such as side-stretching magnetic tweezers + fluorescence. More generally, it would be interesting to see the predictions of a fully developed local denaturation model compared with experiments – for the conditions reported here, the authors dismiss this kind of model on the basis of the AT-tract shuffling experiment, and do not consider it further. Although additional data would be ideal, and perhaps the authors already have such data. At a minimum, the authors should revise the text to be more cautious about generalizing to negative supercoils, and they should discuss possible differences between negative and positive supercoiling.

6) Can the DNA structures that form plectonemes transition to a change in twist? This question arises in part from a related point or question, which is what the authors envision in terms of plectoneme formation in the context of eukaryotic chromatin? It is difficult to imagine there generally being sufficient "free" DNA to form plectonemes on a chromatinised template in eukaryotic cells. It would be interesting to consider and discuss how much space a plectoneme needs to form.

7) The authors suggest that plectonemes are enriched at TSS. This seems overly speculative unless the authors can include additional information/experiments. For example, the correlation between curved sequences and TSS could be due to promoters evolving curved sequences to increase RNA polymerase binding or activity, rather than plectoneme formation per se. Insertion of a curved sequence upstream of the -35 element has been shown to increase affinity for RNA polymerase in vitro on a linear template in the absence of supercoiling (Nickerson et al., 1995). The authors' argument would be strengthened if they could show plectoneme formation on these sequences in vivo or a relationship between plectoneme formation and RNA polymerase binding/activity in vitro. Unless such data are included (and we recognize it may be beyond the scope of this paper), the authors should at least tone down some statements in this vein, e.g. in the Abstract where it says "…and experimentally verify that plectonemes localize directly upstream…".

8) The experiments here are necessarily performed on DNA with intercalated dye. While the authors argue on the basis of prior work that overall mechanical properties of DNA are minimally affected by the intercalator, can they rule out that sequence-specific observations are affected by the dye? MT tweezers assays in the absence of dye cannot directly report on plectoneme locations, but can detect signatures of strong pinning sequences – could targeted key predictions be tested in this kind of assay? On this point the authors are encouraged to provide new data, if easily obtained, or at least discuss the caveats in a revised manuscript.

9) The authors should discuss the assumptions and approximations involved in predicting plectoneme probabilities solely on the basis of tip-loop energetics. Their model falls short of a complete statistical mechanical treatment that would model partitioning of linking number among plectonemes under imposed constraints across the full molecule, and will for example fail to capture position-dependent effects in which the growth of a plectoneme pinned near the end of the molecule is limited by the available DNA, which can favor the formation of a plectoneme elsewhere in the DNA to absorb more linking number (see Bramarchi, Dittmore, et al., 2018).

10) The authors see no effect of changing the ionic strength on the plectonemic density profile. Can they comment on whether this was expected? Have they analyzed whether ionic strength affects the frequency of observing one vs. multiple plectonemes?

11) In the Abstract, the statement "We… verify that plectonemes localize directly upstream of transcriptional start sites" may easily be misread to imply that this measurement has been made in the context of the chromosome. It would of course increase the impact of the paper if plectonemic pinning and/or functional effects of changing the identified sequences could be detected in cells, but that reasonably lies outside the scope of this paper.

12) For a title, is "DNA sequence encodes the position of plectonemes" more accurate as this is what the authors are measuring?

[Editors' note: further revisions were requested prior to acceptance, as described below.]

Thank you for resubmitting your work entitled "DNA sequence encodes the position of DNA supercoils" for further consideration at *eLife*. Your revised article has been favorably evaluated by Naama Barkai as the Senior Editor, and three reviewers, one of whom is a member of our Board of Reviewing Editors.

The manuscript has been improved but there are some remaining issues that need to be addressed before acceptance, as outlined below in the individual reviews. As you'll see, two of the reviewers both emphasize the need for greater attention to detail in discussing some of the results and providing appropriate qualifications and relevant caveats. Please carefully and fully address all of the issues raised by the reviewers in a revised manuscript.

*Reviewer #1:*

The authors have addressed my earlier comments and have significantly improved the manuscript with the addition of new experimental data and further discussion of their results. As noted below, I feel that there are several points in the manuscript that could potentially lead to misunderstandings for the reader and should be clarified.

1) The authors suggest that intrinsic curvature is the major determinant of plectoneme pinning. However, since formation of negative plectonemes is also dependent on the duplex stability, intrinsic curvature may not be the primary mechanism that determines where negative plectonemes are localized. Unless the authors provide experimental data on negatively supercoiled plectonemes, the authors should be careful with their statements such as "intrinsic curvature over a ~70bp range as the primary factor that determines plectoneme pinning…"

2) The authors suggest that their model fails to predict pinning in SeqA, SeqB, and SeqC because of "an insufficient accuracy in the dinucleotide parameters that we adopted from the literature or because the curvature is influenced by interactions spanning beyond nearest-neighbor nucleotides." It is not clear to me that there are not alternative explanations, such as these pinnings being influenced by sequences that are prone to base-flipping or that are able to stabilize twist, or potentially a combination of all the above effects. I think the authors should be more agnostic about the possible mechanisms and acknowledge in the discussion that there are likely other sequence determinants that regulate plectoneme pinning.

3) Given that the authors' model identifies many false-negative pinning sequences, it seems a bit premature to suggest that organisms where their model does not detect strong pinning sequences in the TSS "rely on sequence-dependent plectoneme positioning to different extents". It seems just as likely that these organisms utilize the same as SeqA, SeqB, and SeqC, which is not captured in the authors' model. The authors' argument would be strengthened if they showed in their ISD experiments that *C. crescentus* or *S. cerevisiae* promoters indeed do not contain pinning sequences. Additionally, are SeqA, SeqB, and SeqC sequences derived from genomic DNA? If so, this would further support the idea that there are additional mechanisms that regulate pinning in vivo.

4) The sentence "Our data instead suggests that plectoneme pinning depends on the specific distribution of bases, and our shuffled poly(A/T) constructs suggest this distribution must be measured over distances greater than tens of nucleotides" should be modified to reflect the authors' subsequent results. As shown later in this manuscript, plectoneme pinning (1) does not depend on a "specific" distribution of bases, instead being dependent on the curvature of DNA, and (2) occurs on a ~70 bp length scale.

5) Subsection “Systematic examination of plectoneme pinning at various putative DNA sequences”, last two paragraphs. In this section, several hypotheses are presented and discarded and it is not clear to the reader what is the "correct" model (not solely dependent on AT-character, poly(A/T), etc.). It would be easier for the reader if the statements such as "seemingly confirming our hypothesis" are eliminated since these statements are immediately contradicted in the text.

*Reviewer #2:*

I am satisfied with the revisions.

*Reviewer #3:*

Abbondanzieri and Dekker and coworkers have improved their manuscript and made their claims easier to evaluate by including experiments on new sequences, presenting additional calculation results, and providing additional context in the text. I favor publication, but I think the authors could do even more to qualify their claims and point out the limitations of the current model – this is an important advance but not a definitive determination of a "plectoneme code".

I appreciate the inclusion of additional model predictions in Figure 3—figure supplement 1. As far as I can tell they still haven't included the complete list we asked for – the variable length AT-rich insertions (series of constructs from Figure 2—figure supplement 1) don't seem to be there, and would be a valuable addition to gauge (albeit anecdotally) the extent of the "false negative" effect seen in SeqA/SeqB/SeqC. The latter false negatives could be emphasized more as a caveat in the text, noting that SeqB was used as the basis of shuffling experiments that the authors relied on to motivate the construction of the model, but in fact the difference between SeqB and its shuffled variants is not captured by the model since the SeqB peak is not predicted.

Clearly, both experimental and theoretical investigation of the negative supercoiling regime is an important future direction for this work. When discussing the limitations of the current model as applied to negative supercoiling, it is important to remind the reader of the biological importance of negative supercoiling, e.g. mesophilic bacterial genomes have strongly net negative superhelical density. When arguing that additional theory is needed to describe sequence-dependent strand separation under tension, it would be appropriate to mention existing theoretical work e.g. from Benham on predicting sites of supercoiling-induced denaturation.

In the text, "a full statistical mechanical modeling of the entire plectonemic structure" should probably be something like "…of the entire DNA molecule" or "…of plectonemic structures distributed across the DNA molecule".

I am surprised that the authors did not expect ionic strength effects – e.g. as noted in other studies low ionic strength can change the distribution of plectonemes, favoring multiple plectonemic domains – but I don't think it is critical to include additional experiments to explore that regime.

---

## [Author Response]

Although the reviewers were generally enthusiastic, there were some issues that arose during review and discussion that should be addressed in a revision. The first of these will require additional experiments. The others could, but in some cases may also be dealt with through changes in the text and presentation of the data:1) While this work shows that AT content is not required for plectoneme formation, the authors' conclusion that the curvature is the predominant driver of supercoil formation is based on a limited number of sequence manipulations, with all insertions tested in a single longer context (template 1) and only one synthetic designed sequence tested after developing the model. A deeper analysis (changing the AT content, length, amount of curvature or eliminating curvature entirely) of their template is important to understand the sequence properties that are necessary for plectoneme formation. Since short 60-90 bp segments are sufficient for supercoil pinning (Kremer et al., 1993) and the authors' predict that a 73 bp region should be sufficient to induce pinning, shorter segments than the ones presented (250-450 bp Figure 3E, 4C) should be directly tested. Working with a GC-rich template, in particular, would improve the generality of the results.

To verify our model further, we have acquired an extensive set of additional data. Accordingly, we now added these supporting data sets with various sequence-curvature combinations, specifically:

1) 75bp-long AT-rich (41%) highly-curved

2) 73bp-long neutral (50-55%) flat sequence

3) 500bp non-curved GC-rich (60-70%) insert

4) 500bp non-curved GC-rich (60-70%) + multiple 73bp-long GC-rich (60-70%) curved inserts

In these extended sets of model predictions and measurements, plectoneme pinning was observed only when the inserted sequence was curved, regardless of their AT/GC contents, confirming that it is the intrinsic curvature, not merely AT or GC percentage, which is the major determinant of plectoneme pinning. Furthermore, our new observation of strong pinning from a single short (~73 bp) highly curved sequence is consistent with our plectoneme tip-loop size estimation.

We included these results in Figure 3F-I and Figure 3—figure supplement 1.

We note that the acquisition of the additional data was done by Dr. Eugene Kim, who accordingly has now been added as a co-author.

2) Related to the point above, the authors state that "the model qualitatively represented the experimental data for all sequences that were tested", but it appears this comparison is only shown for a limited subset of the sequences. Given that the paper is centrally focused on proposing a general model, it would be appropriate to show comparisons with all sequences, to facilitate evaluation of the model's successes and limitations. In particular, it does not seem that model predictions are shown for about a dozen of the sequence variants for which experiments are reported:– template 2– template 1 + seqA, seqB, seqC;– seqB with base shuffling and with AT-tract shuffling (the latter in particular is a central experiment for the paper)– template 1 + variable length insertions (.25, 1, 2, 3, 3.9 kb)

We now provide all the model predictions (including all those sequences mentioned in the above list) in a figure supplement (Figure 3—figure supplement 1). This extensive set largely confirms our statement that the model predictions agree with the experimental data. The full in depth analysis revealed that our model can reproduce most of the measured plectoneme densities, indicating that curvature is the major determinant. However, occasionally, we find that the model is too conservative, i.e., while it does a good job avoiding false positives, it suffers from some false negatives. For completeness, we have mentioned this now in the revised manuscript and provide Figure 3—figure supplement 1 which shows examples where an inserted sequence showed a peak in the experiments, whereas the model failed to predict it, which we attribute to the base-base parameters being insufficiently accurate.

3) The Introduction of the manuscript is somewhat misleading. An existing model in the field is that curved sequences induce supercoil formation (many of these papers are cited in the manuscript, see Kremer et al., 1993, Laundon and Griffith, 1988, Pfannschmidt and Langowski, 1998, Tsen and Levene, 1997, and Pavlicek et al., 2004, among others), however the authors do not acknowledge this. The authors are correct that a major shortcoming of these earlier works is their use of phased AT-tracts in generating a curved sequence, however, the introduction should be written to clarify this for the reader. Some of the references are also misattributed. In the second paragraph of the Introduction, Eslami-Mossallam et al., 2016, Pasi and Lavery, 2016, and Wang et al., 2017, are computational studies not biochemical and structural studies, while Kremer et al., 1993, Laundon and Griffith, 1988, Pfannschmidt and Langowski, 1998, and Tsen and Levene, 1997, used biochemical and structural approaches in addition to in silico modeling.

We appreciate the careful examination of our manuscript and the suggestions. We now rephrased and added a statement to properly credit the earlier works and we corrected the misattributed references (Introduction, second paragraph).

4) How the plectoneme density (e.g., Figure 1D) is calculated was buried in the Materials and methods and the prior paper from this group, making it a little difficult to understand what was done. Given the centrality of plectoneme density plots, it would be useful for the definition, calculation procedure, and motivation for choosing this metric to be explained clearly up front. The Materials and methods mention the calculation of the size of each plectoneme (how much writhe is constrained in the molecule at each position) but it is unclear whether the plectoneme size is ever used in any of the reported analysis. Relatedly, how do AT-content, curvature, and other sequence features affect plectoneme formation/termination kinetics and the amount of writhe per plectoneme? Finally, can the authors quantify sequence effects on plectoneme "slithering" (diffusion)? They mention that the model predicts the expected degree of roughness to explain plectoneme diffusion rates, but do not comment on whether sequence differences cause predictable changes to these rates.

We have revised Results and Materials and methods to provide a clearer description on how the plectoneme density is defined (see for example, subsection “DNA sequence favors plectoneme localization at certain spots along supercoiled DNA”, first paragraph, subsection “Data analysis”).

Furthermore, we have removed the statement on the plectoneme size as it is not essential for the data analysis and results in this manuscript.

Finally, we note that our physical model is based on the intrinsic curvature and bending energy which essentially are equilibrium properties. The plectoneme dynamics is an interesting subject in itself (see e.g. our previous paper Van Loenhout et al., 2012), but it is beyond the interest of this manuscript. Henceforth, to keep the paper concise and focused, we did not include the plectoneme nucleation, termination, and diffusion kinetics, as the kinetic data did not give any additional insight into how plectonemes pin to specific sequences on average. This is now noted more clearly at the end of the subsection “DNA sequence favors plectoneme localization at certain spots along supercoiled DNA”.

5) The experiments here are performed in the presence of positive supercoiling. There could be important differences under negative supercoiling, relevant to biological contexts. For example, the authors present evidence for intrinsic curvature as a dominant contributor to sequence-dependent plectonemic pinning. An alternative proposal (Matek et al.) is that duplex stability plays an important role, via the formation of plectoneme tip bubbles. If these occur, they are expected to occur more readily in negatively supercoiled DNA, which favors strand separation. Ideally, the authors would provide any evidence addressing whether plectoneme pinning occurs at the same locations/with the same sequence determinants for positively and negatively supercoiled DNA. Clearly the high-throughput assay used here can only interrogate positively supercoiled DNA, so this would require a lower-throughput assay such as side-stretching magnetic tweezers + fluorescence. More generally, it would be interesting to see the predictions of a fully developed local denaturation model compared with experiments – for the conditions reported here, the authors dismiss this kind of model on the basis of the AT-tract shuffling experiment, and do not consider it further. Although additional data would be ideal, and perhaps the authors already have such data. At a minimum, the authors should revise the text to be more cautious about generalizing to negative supercoils, and they should discuss possible differences between negative and positive supercoiling.

We have revised our text to clarify the limits of our data and model regarding the negative supercoiling (Discussion, eighth paragraph). We can clarify our focus on positive supercoiling as follows:

1) We attempted measurements with the side-stretching magnetic tweezers/fluorescence setup in the past three years but unfortunately encountered technical issues with that which prevented experiments such as those suggested above, and this made us switch to the high-throughput assay employed in this paper.

2) Technically, addressing negative supercoiling is more challenging than positive supercoiling, which is connected to the nature of our assay where the addition of SxO induces positive supercoiling. In the technical paper on the method that we published earlier (Ganji et al., 2016), we reported some data for the plectoneme density for negatively supercoiled DNA (upon removing SxO from pre-stained DNA molecules). While these results were roughly consistent with the data for positive supercoiling, they were noisier and less reproducible from molecule to molecule.

3) In the case of negative supercoils, plectoneme pinning can, as correctly noted by the reviewers, be the convoluted result of curvature and local melting, and separating these two phenomena is not straightforward. As an additional complication, local DNA melting strongly depends on the tension across the DNA (i.e. the end-to-end length in our experiment), which would lead to slightly different plectoneme density profiles from molecule to molecule. Therefore, we limited this study to the pinning of pure plectonemes as set by the flexibility and curvature, which can be best demonstrated in the regime of positive supercoiling.

4) Finally, we note that torque-induced melting is quite different to thermal melting (see for example Vlijm et al., 2015). To our knowledge, it is not yet possible to predict torque-induced melting of DNA directly from sequence. Hence, we did not include the effect of melting in our physical model.

We have revised the text to indicate some of these points and discuss the possible differences between negative and positive supercoiling.

6) Can the DNA structures that form plectonemes transition to a change in twist? This question arises in part from a related point or question, which is what the authors envision in terms of plectoneme formation in the context of eukaryotic chromatin? It is difficult to imagine there generally being sufficient "free" DNA to form plectonemes on a chromatinised template in eukaryotic cells. It would be interesting to consider and discuss how much space a plectoneme needs to form.

Our data are taken in the regime above the buckling transition where the internal twist reaches a maximum and further torsion is all absorbed in plectonemes. Accordingly, the plectonemes cannot transition into twist.

As mentioned in the manuscript, the plectoneme formation in eukaryotic chromatin may be more complicated because the DNA is highly covered by histones. However, we like to point out that histones and other DNA binding proteins do not prevent plectoneme formation. In fact, they can even induce plectoneme formation and pinning by making a sharp bend around their binding site. That may be why the eukaryotic genomes do show less of a propensity of recruiting plectonemes in the promotor region (Figure 4F). As far as the size is considered, only a few hundred base pairs are needed to initiate a plectoneme loop (or ‘curl’ to be more precise). Considering that a chromatin region of a highly transcribing gene is relatively ‘open’ for polymerase binding, it is likely that there are long enough protein-free regions.

7) The authors suggest that plectonemes are enriched at TSS. This seems overly speculative unless the authors can include additional information/experiments. For example, the correlation between curved sequences and TSS could be due to promoters evolving curved sequences to increase RNA polymerase binding or activity, rather than plectoneme formation per se. Insertion of a curved sequence upstream of the -35 element has been shown to increase affinity for RNA polymerase in vitro on a linear template in the absence of supercoiling (Nickerson et al., 1995). The authors' argument would be strengthened if they could show plectoneme formation on these sequences in vivo or a relationship between plectoneme formation and RNA polymerase binding/activity in vitro. Unless such data are included (and we recognize it may be beyond the scope of this paper), the authors should at least tone down some statements in this vein, e.g. in the Abstract where it says "…and experimentally verify that plectonemes localize directly upstream…".

We note that (i) our experiments clearly show a good correspondence between local curvature and plectoneme pinning, and (ii) our bioinformatics data show a good correlation between local curvature and TSS sequence. We do not claim that the promoters evolved curved sequences solely for the purpose of plectoneme pinning, but we find the correspondence striking and noteworthy. We also tested TSS sequences and experimentally found the expected pinning at these sites. We feel that, in vivo experiments to disentangle the likely complex relationship between plectoneme formation and RNA polymerase binding and transcription are beyond the current scope of the manuscript.

We have toned down some statements and rephrased part of the Abstract and Discussion (fifth paragraph) to avoid overclaiming.

8) The experiments here are necessarily performed on DNA with intercalated dye. While the authors argue on the basis of prior work that overall mechanical properties of DNA are minimally affected by the intercalator, can they rule out that sequence-specific observations are affected by the dye? MT tweezers assays in the absence of dye cannot directly report on plectoneme locations, but can detect signatures of strong pinning sequences – could targeted key predictions be tested in this kind of assay? On this point the authors are encouraged to provide new data, if easily obtained, or at least discuss the caveats in a revised manuscript.

We have been keenly aware of possible effects of the intercalating dyes on the mechanical properties of DNA, and accordingly we tested this thoroughly. Indeed, as confirmed experimentally in our magnetic tweezer experiments, Sytox Orange does not in any way perturb the formation of plectonemes on DNA. Furthermore, we observe a homogeneous fluorescence intensity on relaxed DNA, irrespective of the GC content and regardless of how strongly plectonemes are pinned at local positions along the DNA (see Figure 1—figure supplement 1A), indicating that Sytox orange does not show any specificity for high plectoneme pinning sequences or any other specific sequences. Finally, data from the assay reported in this paper were identical to those from an orthogonal approach using side-pulling magnetic tweezers (cf. our previous work, Ganji et al., 2016). All these experiments show that the mechanical properties of DNA are minimally affected, if at all, by the SxO dyes.

9) The authors should discuss the assumptions and approximations involved in predicting plectoneme probabilities solely on the basis of tip-loop energetics. Their model falls short of a complete statistical mechanical treatment that would model partitioning of linking number among plectonemes under imposed constraints across the full molecule, and will for example fail to capture position-dependent effects in which the growth of a plectoneme pinned near the end of the molecule is limited by the available DNA, which can favor the formation of a plectoneme elsewhere in the DNA to absorb more linking number (see Bramarchi, Dittmore, et al., 2018).

Our simple physical model is intended to estimate if a given sequence is capable to pin a plectoneme relatively to other sequences, and it does a good job at that. The model is capable of good qualitative predictions and it requires low computational power. Indeed, better predictions may result from a more extensive full statistical mechanical model, but this will require very significant computational power and time. We have added a discussion on the advantages and limits of our model (Discussion, third and fourth paragraphs).

10) The authors see no effect of changing the ionic strength on the plectonemic density profile. Can they comment on whether this was expected? Have they analyzed whether ionic strength affects the frequency of observing one vs. multiple plectonemes?

Our earlier study (van Loenhout et al., 2012) addressed the question how the ionic strength affects plectoneme dynamics. As pointed out above, however, the plectoneme dynamics is not the focus of this study where we instead investigate the average pinning properties. We therefore did not extensively measure the salt dependence, which we did not expect to have large effects on plectoneme pinning.

11) In the Abstract, the statement "We… verify that plectonemes localize directly upstream of transcriptional start sites" may easily be misread to imply that this measurement has been made in the context of the chromosome. It would of course increase the impact of the paper if plectonemic pinning and/or functional effects of changing the identified sequences could be detected in cells, but that reasonably lies outside the scope of this paper.

We have revised the Abstract to avoid such a misreading.

12) For a title, is "DNA sequence encodes the position of plectonemes" more accurate as this is what the authors are measuring?

We appreciate the suggestion. But we prefer to use the original title “DNA sequence encodes the position of DNA supercoils” because the term “supercoil” is better known to the broad audience than “plectonemes”.

[Editors' note: further revisions were requested prior to acceptance, as described below.]

Reviewer #1:The authors have addressed my earlier comments and have significantly improved the manuscript with the addition of new experimental data and further discussion of their results. As noted below, I feel that there are several points in the manuscript that could potentially lead to misunderstandings for the reader and should be clarified.1) The authors suggest that intrinsic curvature is the major determinant of plectoneme pinning. However, since formation of negative plectonemes is also dependent on the duplex stability, intrinsic curvature may not be the primary mechanism that determines where negative plectonemes are localized. Unless the authors provide experimental data on negatively supercoiled plectonemes, the authors should be careful with their statements such as "intrinsic curvature over a ~70bp range as the primary factor that determines plectoneme pinning…"

We agree that we should be careful on this point. Our experiments clearly show that intrinsic curvature is a major determinant of plectoneme pinning for positive supercoiling regime, in which the duplex form of DNA remains intact. This is likely the case as well for negative supercoiling, but the reviewer is correct that things may be more complicated here as local melting may occur. Hence we have rephrased the text to phrase our statement to be more precise (see Discussion, first paragraph).

2) The authors suggest that their model fails to predict pinning in SeqA, SeqB, and SeqC because of "an insufficient accuracy in the dinucleotide parameters that we adopted from the literature or because the curvature is influenced by interactions spanning beyond nearest-neighbor nucleotides." It is not clear to me that there are not alternative explanations, such as these pinnings being influenced by sequences that are prone to base-flipping or that are able to stabilize twist, or potentially a combination of all the above effects. I think the authors should be more agnostic about the possible mechanisms and acknowledge in the discussion that there are likely other sequence determinants that regulate plectoneme pinning.

We thank the reviewer for suggesting the additional mechanisms as potential causes for plectoneme pinning. We now revised the text to not exclude such other possible explanations (subsection “Intrinsic local DNA curvature determines the pinning of supercoiled plectonemes”, second paragraph).

Furthermore, in order to provide the reader with a more quantitative feel for the relative importance of the choice of the dinucleotide parameters, we now added Figure 3—figure supplement 2. This depicts the model predictions for the various sets of dinucleotide parameters that have been reported in the literature. Interestingly, quite pronounced variations can be seen among the results, providing support for our statement that an increased accuracy in the dinucleotide parameters or inclusion of interactions spanning beyond nearest-neighbor nucleotides may in the future improve the model further. Notably we used the Balasubramian data set for our modeling which is most recent and most accurate.

3) Given that the authors' model identifies many false-negative pinning sequences, it seems a bit premature to suggest that organisms where their model does not detect strong pinning sequences in the TSS "rely on sequence-dependent plectoneme positioning to different extents". It seems just as likely that these organisms utilize the same as SeqA, SeqB, and SeqC, which is not captured in the authors' model. The authors' argument would be strengthened if they showed in their ISD experiments that C. crescentus or S. cerevisiae promoters indeed do not contain pinning sequences. Additionally, are SeqA, SeqB, and SeqC sequences derived from genomic DNA? If so, this would further support the idea that there are additional mechanisms that regulate pinning in vivo.

First, we like to point out that our model is remarkably successful for predicting plectoneme pinning on the majority of the sequences examined in the manuscript (15 different sequence inserts plus two ~20kb-long templates) and its deficiency for these few examples does in our view not merit the reviewer’s statement that our model “identifies many false-negative pinning sequences”.

Furthermore, since the genome scan data in Figure 4 are the result from averaging thousands of TSS sites, it is reasonable to conclude that the near-zero pinning probabilities for the *C. crescentus* and *S. cerevisiae* are hardly affected by false negative detection of our model.

4) The sentence "Our data instead suggests that plectoneme pinning depends on the specific distribution of bases, and our shuffled poly(A/T) constructs suggest this distribution must be measured over distances greater than tens of nucleotides" should be modified to reflect the authors' subsequent results. As shown later in this manuscript, plectoneme pinning (1) does not depend on a "specific" distribution of bases, instead being dependent on the curvature of DNA, and (2) occurs on a ~70 bp length scale.

We revised the text for better coherency (subsection “Systematic examination of plectoneme pinning at various putative DNA sequences”, last paragraph).

5) Subsection “Systematic examination of plectoneme pinning at various putative DNA sequences”, last two paragraphs. In this section, several hypotheses are presented and discarded and it is not clear to the reader what is the "correct" model (not solely dependent on AT-character, poly(A/T), etc.). It would be easier for the reader if the statements such as "seemingly confirming our hypothesis" are eliminated since these statements are immediately contradicted in the text.

We admit that the logical flow in the section may come across as a bit confusing, as we explore a number of hypotheses that fail. While we were in fact quite careful in writing this section already, we have now further rephrased the text to avoid such confusion (for example, see subsection “Systematic examination of plectoneme pinning at various putative DNA sequences”, second and third paragraphs).

Reviewer #3:Abbondanzieri and Dekker and coworkers have improved their manuscript and made their claims easier to evaluate by including experiments on new sequences, presenting additional calculation results, and providing additional context in the text. I favor publication, but I think the authors could do even more to qualify their claims and point out the limitations of the current model – this is an important advance but not a definitive determination of a "plectoneme code".I appreciate the inclusion of additional model predictions in Figure 3—figure supplement 1. As far as I can tell they still haven't included the complete list we asked for – the variable length AT-rich insertions (series of constructs from Figure 2—figure supplement 1) don't seem to be there, and would be a valuable addition to gauge (albeit anecdotally) the extent of the "false negative" effect seen in SeqA/SeqB/SeqC. The latter false negatives could be emphasized more as a caveat in the text, noting that SeqB was used as the basis of shuffling experiments that the authors relied on to motivate the construction of the model, but in fact the difference between SeqB and its shuffled variants is not captured by the model since the SeqB peak is not predicted.

Upon this comment of the reviewer, we now added the predictions on the variable length AT-rich insertions in Figure 3—figure supplement 1, and we also added a statement to more clearly address the caveat of false negative predictions (see Discussion, fourth paragraph).

Furthermore, we note that the sequences used in Figure 2—figure supplement 1 are merely lengthened or shortened versions of SeqA, yet the model predicts an increase of the pinning effect by the lengthened AT-region from 0.25kb to 1kb (see Author response image 1). Also please note that the peak is broadened from 3.0 kb to 3.9kb as the AT-rich insert becomes even larger than the smoothing window (1.6kb). Thus, our model does not entirely miss the pinning mechanism, but somewhat underestimates the effect.

Clearly, both experimental and theoretical investigation of the negative supercoiling regime is an important future direction for this work. When discussing the limitations of the current model as applied to negative supercoiling, it is important to remind the reader of the biological importance of negative supercoiling, e.g. mesophilic bacterial genomes have strongly net negative superhelical density. When arguing that additional theory is needed to describe sequence-dependent strand separation under tension, it would be appropriate to mention existing theoretical work e.g. from Benham on predicting sites of supercoiling-induced denaturation.

We agree with the reviewer. We now added a statement regarding this interesting early theoretical work on supercoiling-induced denaturation, which will be of relevance for future studies on negative supercoiling (Discussion, eighth paragraph).

In the text, "a full statistical mechanical modeling of the entire plectonemic structure" should probably be something like "…of the entire DNA molecule" or "…of plectonemic structures distributed across the DNA molecule".

We have revised the text accordingly.

I am surprised that the authors did not expect ionic strength effects – e.g. as noted in other studies low ionic strength can change the distribution of plectonemes, favoring multiple plectonemic domains – but I don't think it is critical to include additional experiments to explore that regime.

As we mentioned in the previous rebuttal, the ionic strength mainly affects the dynamics of supercoils, which will merely induce a broadening in the steady-state plectoneme density plots. Hence, the density profiles will not change essentially.